# GMV: A Unified and Efficient Graph Multi-View Learning Framework

**Qipeng Zhu**[1]*, **Jie Chen**[2]*, **Jian Pu**[3]†, **Junping Zhang**[1]†
[1]Shanghai Key Laboratory of Intelligent Information Processing,
College of Computer Science and Artificial Intelligence, Fudan University
[2] College of Computer and Data Science, Fuzhou University
[3]Institute of Science and Technology for Brain-Inspired Intelligence, Fudan University
qpzhu23@m.fudan.edu.cn, jiechen202@fzu.edu.cn,
{jpzhang,jianpu}@fudan.edu.cn

## Abstract

Graph Neural Networks (GNNs) are pivotal in graph classification but often struggle with generalization and overfitting. We introduce a unified and efficient Graph Multi-View (GMV) learning framework that integrates multi-view learning into GNNs to enhance robustness and efficiency. Leveraging the lottery ticket hypothesis, GMV activates diverse sub-networks within a single GNN through a novel training pipeline, which includes mixed-view generation, and multi-view decomposition and learning. This approach simultaneously broadens "views" from the data, model, and optimization perspectives during training to enhance the generalization capabilities of GNNs. During inference, GMV only incorporates additional prediction heads into standard GNNs, thereby achieving multi-view learning at minimal cost. Our experiments demonstrate that GMV surpasses other augmentation and ensemble techniques for GNNs and Graph Transformers across various graph classification scenarios. The open source code can be found in https://github.com/smurf-1119/GMV.

## 1 Introduction

Graph Neural Networks (GNNs) have emerged as a powerful tool for graph classification tasks, attracting considerable attention. Despite their success, GNNs struggle with generalization and overfitting due to the complex nature of graph structures and the limited availability of labeled graph data [1, 2]. As shown in Fig 1, simply increasing the parameters of GNNs does not consistently enhance their performance [3]. A promising solution lies in multi-view learning, which enables models to extract diverse representations by aggregating complementary perspectives of data [4, 5]. By forcing models to reconcile differences across views, multi-view learning offers a fundamental insight of diversity for enhancing model generalization.

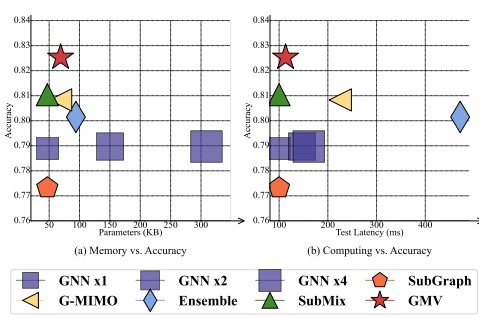

Figure 1: Accuracy vs. memory/speed. Specifically, "GNN x∗" represent different sizes of GNNs.

---

*Qipeng Zhu and Jie Chen are co-first authors.
†Junping Zhang and Jian Pu are corresponding authors.

39th Conference on Neural Information Processing Systems (NeurIPS 2025).

Existing graph learning strategies implicitly leverage multi-view principles but remain suboptimal. Graph data augmentation (e.g., DropEdge [6], S-Mixup [7]) diversifies input views via edge removal or graph interpolation, acting as "data-view" expansions. However, these methods often degrade structural integrity (e.g., random edge dropping disrupts critical topological hierarchies [8]), limiting their effectiveness on structured graphs. Ensemble learning [9, 10, 11] achieves "model-view" diversity by training multiple GNNs, but at the cost of significant computational overhead as illustrated in Fig 1. The need for separate forward passes across networks renders these methods infeasible for large graphs. These strategies treat data and model views in isolation, failing to exploit the synergistic power of multi-view learning.

In this paper, we introduce a unified and efficient **G**raph **M**ulti-**V**iew (GMV) learning framework. GMV is model-agnostic and expands views from three complementary perspectives—data, model, and optimization—to activate diverse sub-networks within a single GNN. Inspired by the lottery ticket hypothesis [12], where neural networks contain latent sub-networks with comparable performance to the full model, we aim to overcome the challenge that standard supervised training fails to activate such diversity [13, 4]. Specifically, we design a novel training pipeline integrating mixed-view generation and multi-view decomposition and learning.

During training, GMV employs a three-fold coherent strategy to unify multi-view learning. *From a data perspective*, we propose structure enhanced subgraph mixing, which samples two subgraphs that preserve both the topological structure and semantic nodes to generate mixed graph views. This mixed view contains the multi-view knowledge and addresses the structural loss in prior augmentations. *From a model perspective*, we introduce a lightweight dual-output prediction head to explicitly activate two sub-networks within any single GNN and Graph Transformer (GT). This design enables parallel encoding of mixed views and multi-view decomposition in one forward pass, eliminating the multi-model overhead of ensemble methods while preserving representation diversity. *From a optimization perspective*, we design multi-view and mixed-view loss functions. These two losses collectively supervise view-specific predictions and activate sub-networks to learn diverse multi-view representations. During inference, GMV processes standard graph input and simply averages dual-head outputs with single-model efficiency. By unifying data, model, and optimization perspectives of multi-view learning, GMV provides a generalizable solution for GNNs and GTs. As illustrated in Fig 1, GMV achieves the best trade-off between overhead and accuracy.

Our contribution can be summarized as follows: 1) We introduce GMV, a unified and efficient multi-view learning framework that enhances the robustness and generalization of both GNNs and GTs in graph classification tasks. 2) We propose new structure-enhanced subgraph mixing techniques, accompanied by multi-view and mixed-view loss, to encourage models to learn from diverse graph views. 3) Our comprehensive experiments evaluate the efficacy, robustness, and generalization of GMV. GMV significantly improves GNNs/GTs and achieves state-of-the-art results compared to various graph augmentation and graph ensembling methods.

## 2   Related Work

**Graph Neural Network.** Graph Neural Networks (GNNs) leverage the message passing mechanism [14, 15] to aggregate and update node representations for graph data processing [16, 17, 18]. The Graph Convolutional Network (GCN) [19] uniformly aggregates neighbor messages to update node embeddings. GraphSAGE [20] introduces subgraph sampling with diverse aggregation methods for adaptive representations. The Graph Isomorphism Network (GIN) [21] further refines this by capturing graph isomorphism, enhancing model sensitivity to graph topology. Moreover, combining the GNN with transformer architecture, such as Graphomer [22] and GraphGPS [23], has also emerged in graph learning fields.

**Multi-view Learning.** In computer vision, multi-view data, typically derived from various perspectives with shared high-level semantics, has become a crucial data type [24]. Asif et al. [4] apply multi-view learning theory to multi-class classification, suggesting that each image has an inherent "multi-view" structure, where these "multi-view" structures correspond to multiple data features that can help deep neural networks in accurate classification. They demonstrate how multi-view learning can improve both the generalization and robustness of deep neural networks. While several multi-view learning strategies [25, 26, 27, 28, 29] have been proposed for graph tasks, their application to supervised graph classification remains challenging due to differences in task objectives

and data characteristics. For example, Yuan et al. [27] generate node feature views for both labeled and unlabeled nodes in node classification, whereas Liu et al. [28] generate views based on pairs of positive and unlabeled graphs in graph classification. Both focus on semi-supervised learning. Compared to image classification, generating mixed-views that preserve both structural and semantic information in graph classification is more difficult. In this paper, we propose generating mixed-views to activate dual sub-networks within GNNs, enhancing multi-view learning capabilities from the data, model, and optimization perspectives.

**Graph Augmentation.** We conceptualize graph augmentation as a specialized form of multi-view learning, aimed at expanding graph datasets through modifications. One approach involves randomly modifying the original graph while assuming the label remains unchanged, such as DropNode [30], DropEdge [6], and Subgraph [31]. However, the simplicity of these operations often limits the diversity of the resulting graph views and may introduce noise. Other approaches integrate mixup techniques [32] into graph classification. For example, S-Mixup [7] aligns pairs of graphs using a soft alignment matrix derived from a trained Graph Matching Network (GMN), followed by linear interpolation of the aligned graphs. Nevertheless, the complexity and resource demands of training an effective GMN often lead to suboptimal performance due to inadequate mapping. Techniques like SubMix [33] and GraphTransplant [34] connect subgraphs sampled from different graphs to facilitate model-agnostic graph augmentation. However, these methods do not fully exploit the sub-views of graphs and often neglect structural information. In contrast, GMV effectively integrates structure-enhanced sub-views to generate mixed views, while utilizing a multi-view decomposition and learning pipeline to extract diverse view representations.

**Ensemble Learning.** Ensemble learning [9, 4, 35] aims to improve the robustness and generalization of a single model by combining the outputs of multiple models. This approach, however, comes with high computational and memory demands. The Lottery Ticket Hypothesis [12, 36] posits that dense neural networks contain sparse subnetworks ("winning tickets") capable of achieving comparable performance when trained in isolation, which suggests the possibility of ensemble learning with these subnetworks. In the realm of image classification, MIMO [13] introduces multi-input multi-output techniques to ensemble sub-networks within a single convolutional neural network. Despite these advancements, applying ensemble learning effectively to Graph Neural Networks (GNNs) remains a challenge, primarily due to the arbitrary sizes of graphs. G-MIMO [37] addresses this by implementing graph multi-input and multi-output schemes, adding multiple parallel graph encoders and decoders. However, this approach complicates the forward passing process in GNNs and struggles with limited graph views. In contrast, our proposed method, GMV, minimizes transformations for GNNs and achieves efficient ensembling through a single forward pass, efficiently enhancing the multi-view learning capability.

# 3 Method

To enhance the robustness and generalization of GNNs through multi-view graph learning, we simultaneously increase the diversity of input graph views and the multi-view learning capabilities of GNNs. As illustrated in Figure 2, we first outline the process of mixed-view generation. Then, we introduce details of mixed-view decomposition and multi-view learning, which activate dual sub-networks within a single GNN for efficient ensemble.

## 3.1 Preliminaries

An undirected graph $\mathcal{G} = < \mathcal{V}, \mathcal{E}, \mathbf{A}, \mathbf{X} >$, where $\mathcal{V} = \{v_i | 1 \leq i \leq n\}$ represents the set of nodes, and $\mathcal{E} = \{e_{ij} | v_i \in \mathcal{V} \wedge v_j \in \mathcal{V} \wedge v_i \text{ is connected to } v_j\}$ is the set of edges. The matrix $\mathbf{X} \in \mathbb{R}^{n \times d}$ contains the node features, while $\mathbf{A} \in \{0, 1\}^{n \times n}$ is the adjacency matrix where $\mathbf{A}_{ij} = 1$ if nodes $v_i$ and $v_j$ are connected. The degree matrix $\mathbf{D} \in \mathbb{R}^{n \times n}$ has entries $\mathbf{D}_{ii} = \sum_j \mathbf{A}_{ij}$ on the diagonal, with $\mathbf{D}_{ij} = 0$ for $i \neq j$. Each node $v_i \in \mathcal{V}$ has a neighborhood set, denoted $\mathcal{N}(v_i) = \{v_j | v_i \text{ is connected to } v_j \wedge v_j \in \mathcal{V}\}$. For graph classification, a collection of $n$ undirected graphs is represented as $\mathbb{G} = \{(\mathcal{G}_t, \mathbf{y}_t)\}_{t=1}^{n}$, where $\mathbf{y}_t \in \{0, 1, \ldots, C-1\}$ denotes the label for each graph $\mathcal{G}_t$, and $C$ is the number of classes.

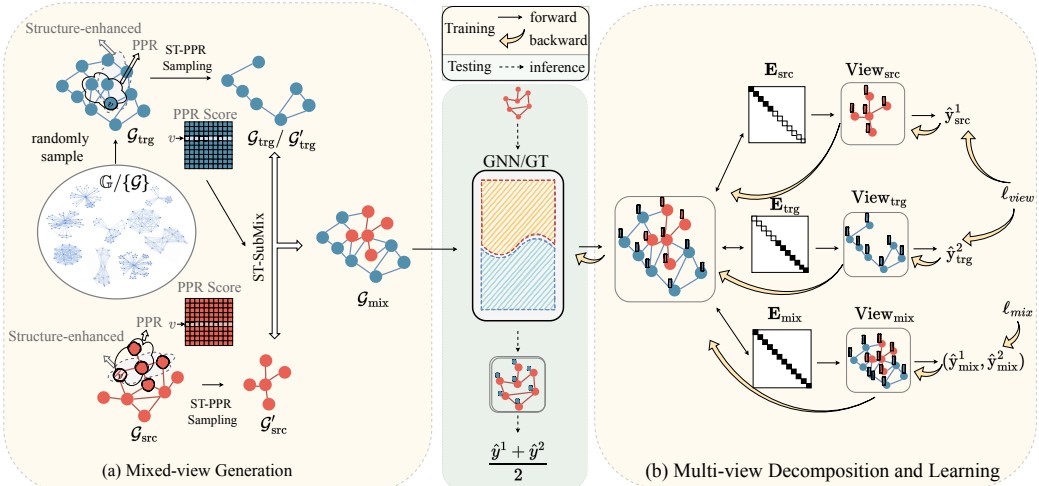

Figure 2: (a) For data perspective, GMV connects two structure enhanced subgraphs to generate the mixed-view. (b) For model perspective, GMV employs dual sub-networks in GNN/GT to gain diverse view representations, denoted as multi-view decomposition. For optimization perspective, we design the multi-view learning process with multi-view ($\ell_{\text{view}}$) and mixed-view loss ($\ell_{\text{mix}}$) to optimize dual sub-networks. When testing, GMV simply averages two predictions of dual sub-networks in GNN/GT as the final output.

## 3.2 Mixed-view Generation

From a data perspective, we explore how to integrate diverse views from different graphs into a single graph, allowing the GNN to process them concurrently and activating sub-networks to learn multi-view representations. Unlike previous graph augmentations [6, 7, 31], our method explicitly considers the critical structural information [38] to generate a mixed graph view. This is achieved through a structure-enhanced subgraph sampling, followed by structure-enhanced subgraph mixing.

### 3.2.1 Structure Enhanced Subgraph Sampling

Unlike random corruption of graphs [6, 30], sampling subgraphs preserves more semantic information [39]. We employ subgraph sampling methods to construct richer views. A key challenge is exploring various subgraphs that encapsulate the most crucial semantic and structural information. Compared to randomly sampling [40], subgraph sampling methods based on Personalized PageRank (PPR) [41] and Determinantal Point Processes (DPP) [42, 43] can enhance the performance of GNNs without altering their architectures. However, the PPR-based method does not explicitly preserve the structure of the original graph, while the DPP-based method may overlook some key nodes due to its limited search scope. Considering that topology information effectively preserves label information during subgraph sampling [44], we propose a novel **ST**ructure Enhanced **PPR** subgraph sampling method (**ST-PPR**), which considers both key nodes and structural information.

The specific process is outlined in Algorithm 1. We first pick a random root node $v$ from the graph $\mathcal{G}$. We consider both structural and semantic information of $\mathcal{G}$ by merging different node candidate sets [44]. Depth-First-Search (DFS) algorithm and Breath-First-Search (BFS) algorithm [45] can easily extract the original topology structure of $\mathcal{G}$. And the PPR algorithm considers semantic information by iteratively calculating the importance score of every node in $\mathcal{G}$ [33]. Therefore, we respectively use DFS, BFS and PPR methods to gain sampling node set $\{\mathcal{V}_{\text{BFS}}, \mathcal{V}_{\text{DFS}}, \mathcal{V}_{\text{PPR}}\}$ from $\mathcal{G}$. We set $w$ as the maximum searching steps for DFS and BFS algorithms. To preserve those important nodes, we calculate the affinity personalized pagerank score matrix $\mathbf{S}_{\text{PPR}}$ [41] as follows:

$$\mathbf{S}_{\text{PPR}} = \sum_{r=0}^{\infty} \beta(1-\beta)^r \left( \mathbf{D}^{-1/2}(\mathbf{A} + \mathbf{I})\mathbf{D}^{-1/2} \right)^r, \qquad (1)$$

where $\mathbf{D}$ and $\mathbf{A}$ respectively is the degree matrix and the adjancy matrix of $\mathcal{G}$ and $\mathbf{I}$ is the identity matrix. We set teleport probability $\beta$ as 0.15 and affinity scores of nodes with respect to node $v$

are contained in $\mathbf{S}_{\mathrm{PPR}}[:,v]$. Then we sort nodes in $\mathcal{V}$ following the scores $\mathbf{S}_{\mathrm{PPR}}[:,v]$ and select top $s_{\mathrm{PPR}}$ nodes to get the node set $\mathcal{V}'_{\mathrm{PPR}}$. And $\mathcal{V}'_{\mathrm{BFS}}$ and $\mathcal{V}'_{\mathrm{DFS}}$ both contain $s_2$ nodes respectively sampled from $\mathcal{V}_{\mathrm{DFS}}$ and $\mathcal{V}_{\mathrm{BFS}}$. We merge three node sets $\{\mathcal{V}'_{\mathrm{PPR}}, \mathcal{V}'_{\mathrm{DFS}}, \mathcal{V}'_{\mathrm{BFS}}\}$ and reorder nodes by $\mathbf{S}_{\mathrm{PPR}}[:,v]$ to obtain $\mathcal{V}'$.

---

**Algorithm 1** Structure Enhanced PPR Subgraph Sampling

---

**Input**: Graph $\mathcal{G} =< \mathcal{V}, \mathcal{E}, \mathbf{A}, \mathbf{X} >$, augmentation ratio of $p \in (0,1)$, structure augmentation ratio of $q$, number of walks $w$
**Output**: Ordered node set $\mathcal{V}'$

1: $v \leftarrow$ pick a random root node from $\mathcal{G}$.
2: $s_{\mathrm{PPR}} \leftarrow$ sample size is $\max\{\mathbb{U}(0,p) \cdot |\mathcal{G}| - q, 0\}$
3: $s_2 \leftarrow$ sample size is $\lfloor (p \cdot |\mathcal{G}| - s_{\mathrm{PPR}})/2 \rfloor$
4: $\mathbf{S}_{\mathrm{PPR}} \leftarrow$ compute score by $\mathrm{PPR}(\mathcal{G}, r)$
5: $\mathcal{V}_{\mathrm{PPR}} \leftarrow \mathrm{Sort}(\mathcal{V}, \mathbf{S}_{\mathrm{PPR}}[:,v]), \mathcal{V}'_{\mathrm{PPR}} \leftarrow \mathcal{V}_{\mathrm{PPR}}[: s_{\mathrm{PPR}}]$
6: $\mathcal{V}_{\mathrm{DFS}} \leftarrow \mathrm{DFS}(\mathcal{G}, v, w), \mathcal{V}'_{\mathrm{DFS}} \leftarrow \mathrm{Sample}(\mathcal{V}_{\mathrm{DFS}}, s_2)$
7: $\mathcal{V}_{\mathrm{BFS}} \leftarrow \mathrm{BFS}(\mathcal{G}, v, w), \mathcal{V}'_{\mathrm{BFS}} \leftarrow \mathrm{Sample}(\mathcal{V}_{\mathrm{BFS}}, s_2)$
8: $\mathcal{V}' \leftarrow$ merge $\{\mathcal{V}'_{\mathrm{PPR}}, \mathcal{V}'_{\mathrm{DFS}}, \mathcal{V}'_{\mathrm{BFS}}\}$ and sort them by $\mathbf{S}_{\mathrm{PPR}}$

---

Combining PPR, BFS, and DFS, the sampled subgraphs covers global hubs, local communities, and long-range paths. This ensures comprehensive feature extraction including global topology , hierarchical transitions and local communities, which boosts the performance of GNNs. The proof is stated in Appendix 6.1.

### 3.2.2 Structure Enhanced Subgraph Mixing

To enable GNNs to effectively process diverse views simultaneously for multi-view learning, we integrate these views of diverse sub-graphs into a single mixed-graph. Inspired by SubMix [33], we propose a **ST**ructure-enhanced **Sub**graph **Mix**ing method (ST-SubMix), which connects two subgraphs according to a node mapping algorithm based on $\mathbf{S}_{\mathrm{PPR}}$. Compared to SubMix, ST-SubMix connects two structure-enhanced subgraph views, thereby preserving more structure and label information from original graphs. The specific process is detailed in Algorithm 2.

Given a source graph (a primary training sample within a given batch), $\mathcal{G}_{\mathrm{src}}$, we randomly sample a target graph (another graph from the same training batch), $\mathcal{G}_{\mathrm{trg}}$ from $\mathbb{G}/\{\mathcal{G}_{\mathrm{src}}\}$. We connect two subgraphs sampled from them to generate $\mathcal{G}_{\mathrm{mix}}$. According to Algorithm 1, we gain $\mathcal{V}'_{\mathrm{src}}$ and $\mathcal{V}'_{\mathrm{trg}}$ respectively sampled from $\mathcal{V}_{\mathrm{src}}$ and $\mathcal{V}_{\mathrm{trg}}$. To ensure the equality of sizes between $\mathcal{V}'_{\mathrm{src}}$ and $\mathcal{V}'_{\mathrm{trg}}$, we let $s = \min\{\mathcal{V}_{\mathrm{src}}, \mathcal{V}_{\mathrm{trg}}\}$. To efficiently mapping two node sets, we make the one-to-one mapping from $\mathcal{V}'_{\mathrm{src}}$ to $\mathcal{V}'_{\mathrm{trg}}$. As shown in Fig 2, we connect $\mathcal{G}'_{\mathrm{src}}$ and $\mathcal{G}_{\mathrm{trg}}/\mathcal{G}'_{\mathrm{trg}}$, which ensures the size distribution of graphs keeping the same as the original distribution [33]. Specifically, we replace the subgraph $\mathcal{G}'_{\mathrm{src}}$ in the graph $\mathcal{G}_{\mathrm{src}}$ with the subgraph $\mathcal{G}'_{\mathrm{trg}}$. To represent the label of the mixed-view, we calculate the confidence of labels of two graphs. As described in Equation (2), the confidence is measured by the count of edge sets within it:

$$w_{\mathrm{src}} = 1 - |\mathcal{E}'_{\mathrm{trg}}|/|\mathcal{E}_{\mathrm{mix}}|, w_{\mathrm{trg}} = |\mathcal{E}'_{\mathrm{trg}}|/|\mathcal{E}_{\mathrm{mix}}|. \tag{2}$$

The procedure in Algorithm 1 outlines a subgraph interpolation method for graph augmentation. It first establishes a canonical node correspondence between a source ($\mathcal{G}_{\mathrm{src}}$) and target ($\mathcal{G}_{\mathrm{trg}}$) graph by ordering their respective nodes via Personalized PageRank (PPR) scores, following the SubMix methodology. This alignment guides the replacement of a target subgraph with its source counterpart to generate a mixed-view graph. For downstream representation decomposition, two binary assignment matrices, $\mathbf{E}\mathrm{src}$ and $\mathbf{E}\mathrm{trg}$, are constructed. Each row is a one-hot vector indicating if a node in the mixed graph originates from the source or target. The property $\mathbf{I} = \mathbf{E}_{\mathrm{src}} + \mathbf{E}_{\mathrm{trg}}$ ensures a disjoint partition of the node set, which is used to separate the view-specific representations from the mixed-view output.

**Algorithm 2** Structure Enhanced Subgraph Mixing

---

**Input**: Graph $\mathcal{G}_{\text{src}} = <\mathcal{V}_{\text{src}}, \mathcal{E}_{\text{src}}, \mathbf{A}_{\text{src}}, \mathbf{X}_{\text{src}}>$, Graph $\mathcal{G}_{\text{trg}} = <\mathcal{V}_{\text{trg}}, \mathcal{E}_{\text{trg}}, \mathbf{A}_{\text{trg}}, \mathbf{X}_{\text{trg}}>$
**Output**: Mixed graph $\mathcal{G}_{\text{mix}} = <\mathcal{V}_{\text{mix}}, \mathcal{E}_{\text{mix}}, \mathbf{A}_{\text{mix}}, \mathbf{X}_{\text{mix}}>$, assignment matrices $\mathbf{E}_{\text{src}}, \mathbf{E}_{\text{trg}}$, confidence of labels of two graphs $w_{\text{src}}, w_{\text{trg}}$

 1: $\mathcal{V}'_{\text{src}}, \mathcal{V}'_{\text{trg}} \leftarrow$ sample subgraphs respectively from $\mathcal{G}_{\text{src}}, \mathcal{G}_{\text{trg}}$ ▷ ST-PPR based Subgraph Sampling 1
 2: $s \leftarrow \min\{|\mathcal{V}'_{\text{src}}|, |\mathcal{V}'_{\text{trg}}|\}$
 3: $\mathcal{V}'_{\text{src}} \leftarrow \mathcal{V}'_{\text{src}}[: s], \mathcal{V}'_{\text{trg}} \leftarrow \mathcal{V}'_{\text{trg}}[: s]$
 4: $\phi \leftarrow$ Make the one-to-one mapping from $\mathcal{V}'_{\text{src}}$ to $\mathcal{V}'_{\text{trg}}$
 5: $\mathcal{E}'_{\text{trg}} \leftarrow \{(u, v) | (u, v) \in \mathcal{E}_{\text{trg}} \wedge \neg (u \in \mathcal{V}'_{\text{trg}} \wedge v \in \mathcal{V}'_{\text{trg}})\}$
 6: $\mathcal{E}'_{\text{src}} \leftarrow \{(\phi(u), \phi(v)) | (u, v) \in \mathcal{E}_{\text{src}} \wedge (u \in \mathcal{V}'_{\text{src}} \wedge v \in \mathcal{V}'_{\text{src}})\}$
 7: $\mathcal{V}_{\text{mix}}, \mathcal{E}_{\text{mix}}, \mathbf{X}_{\text{mix}} \leftarrow \mathcal{V}_{\text{trg}}, \mathcal{E}'_{\text{src}} \cup \mathcal{E}'_{\text{trg}}, \mathbf{X}_{\text{trg}}$
 8: $\mathbf{X}_{\text{mix}}[\phi(\mathcal{V}'_{\text{src}})] \leftarrow \mathbf{X}_{\text{src}}[\mathcal{V}'_{\text{src}}]$
 9: $\mathbf{A}_{\text{mix}} \leftarrow$ densify the edge set $\mathcal{E}_{\text{mix}}$
10: $w_{\text{src}}, w_{\text{trg}} \leftarrow 1 - |\mathcal{E}'_{\text{trg}}|/|\mathcal{E}_{\text{mix}}|, |\mathcal{E}'_{\text{trg}}|/|\mathcal{E}_{\text{mix}}|$
11: $\mathbf{E}_{\text{src}}, \mathbf{E}_{\text{trg}} \leftarrow$ use one hot vectors to record nodes in $\mathcal{V}_{\text{mix}}$ originated from $\mathcal{V}'_{\text{src}}$ and $\mathcal{V}'_{\text{trg}}$

---

### 3.3 Multi-view Decomposition and Learning

From a model perspective, ensembles of diverse neural networks can be seen as learning varied representations of views, thereby improving generalization [4]. However, combining several networks with multiple forward passes leads to high computational costs.

We introduce an innovative pipeline for multi-view decomposition and learning, which activates two sub-networks within a single GNN with minimal computational overhead. During training, we utilize a dual-output predictor with mixed and multi-view loss functions to ensure the learning of multi-view from an optimization perspective.

#### 3.3.1 Mixed-view Encoding

We utilize standard GNNs to encode the mixed-view graph $\mathcal{G}_{\text{mix}}$, which typically leverage repeated message passing process. The process of the $l$-th message passing $\text{MPNN}_l(\cdot)$ in GNNs is formulated as follows:

$$\mathbf{H}_{\text{mix}}^{(l)} = \text{MPNN}_l\left(\mathbf{H}_{\text{mix}}^{(l-1)}, \mathbf{A}_{\text{mix}}\right), \tag{3}$$

where $\mathbf{H}^{(l)}$ denotes the $l$-th layer output of our GMV. We consider the node features $\mathbf{X}_{\text{mix}}$ of $\mathcal{G}_{\text{mix}}$ as $\mathbf{H}_{\text{mix}}^{(0)}$ during training. The output of the mixed-view encoder in GNN as $\mathbf{H}_{\text{mix}}^{(L)}$.

Moreover, we also consider GraphGPS [23] as the shared graph transformer backbone. For each layer of GraphGPS, it consists of three components, including $\text{MPNN}_l(\cdot)$, $\text{GlobalAttn}_l(\cdot)$ and $\text{MLP}_l(\cdot)$. Therefore, the process can be decribed as follows:

$$\mathbf{H}_{\text{mix}}^{(l)} \leftarrow \text{MLP}_l(\text{GlobalAttn}_l(\mathbf{H}_{\text{mix}}^{(l)})). \tag{4}$$

#### 3.3.2 Multi-view Decomposition

Diverse views offer greater evidence for GNN to classify graphs. Given mixed-view representation $\mathbf{H}_{\text{mix}}^{(L)}$, we introduce a Multi-View Decomposition (MVD) to obtain three view representations, denoted as $\{\text{View}_i \mid i \in \{\text{src}, \text{trg}, \text{mix}\}\}$. The MVD can be formulated as follows:

$$\mathbf{View}_i = \mathbf{E}_i \mathbf{H}_{\text{mix}}^{(L)}, \tag{5}$$

where $\{\mathbf{E}_i \mid i \in \{\text{src}, \text{trg}, \text{mix}\}\}$ are assignment matrices, which are calculated in Sec 3.2.2. Then, we utilize a common mean pooling layer [21, 46, 47], denoted as $\text{Pool}(\cdot)$, to respectively readout graph representations of diverse views, i.e., $\{\mathbf{p}_i \mid i \in \{\text{src}, \text{trg}, \text{mix}\}\}$:

$$\mathbf{p}_i = \text{Pool}(\mathbf{View}_i). \tag{6}$$

| | Method | IMDBB | PROTEINS | NCI1 | NCI109 | REDDITB | IMDBM | REDDIT-M5 | COLLAB |
|---|---|---|---|---|---|---|---|---|---|
| | #graphs | 1000 | 1113 | 4110 | 4127 | 2000 | 1500 | 4999 | 5000 |
| | #classes | 2 | 2 | 2 | 2 | 2 | 2 | 3 | 5 |
| | #avg nodes | 19.8 | 39.1 | 29.9 | 29.7 | 429.6 | 13.0 | 508.5 | 74.5 |
| | #avg edges | 96.5 | 72.8 | 32.3 | 32.1 | 497.8 | 65.9 | 594.9 | 2457.2 |
| | Vanilla | 72.30±4.34 | 72.15±3.75 | 72.38±2.15 | 70.27±2.68 | 87.60±2.55 | 49.00±3.96 | 50.83±3.92 | 81.16±1.72 |
| | DropEdge | 72.10±4.21 | 73.41±4.25 | 73.94±2.73 | 67.19±2.42 | 89.25±3.03 | 48.87±3.07 | 50.29±2.21 | 81.56±0.88 |
| | DropNode | 73.30±2.76 | 72.69±4.25 | 73.07±2.96 | 69.76±1.91 | 88.45±2.64 | 49.93±3.56 | 53.73±2.98 | 81.50±2.32 |
| | Subgraph | 72.70±5.16 | 73.05±3.70 | 72.60±2.37 | 69.13±2.72 | 89.30±2.61 | 49.27±3.83 | 50.09±3.45 | 81.42±1.21 |
| | M-Mixup | 73.70±4.12 | 72.15±4.26 | 65.16±2.48 | 62.92±2.15 | 87.60±3.67 | 49.80±3.90 | 48.91±2.08 | 75.58±1.72 |
| GCN | G-Mixup | 73.20±5.60 | 71.18±3.32 | 72.75±1.72 | 72.23±2.50 | 86.85±2.30 | 49.33±3.67 | 51.77±1.42 | 81.17±1.70 |
| | Submix | 73.80±3.57 | 73.50±5.38 | 75.40±2.18 | 72.91±8.25 | 87.90±3.92 | 49.00±3.75 | 53.11±2.03 | 82.62±2.12 |
| | S-Mixup | 72.50±2.20 | 72.42±4.19 | 67.27±2.33 | 69.57±2.56 | 88.50±1.24 | 49.93±3.51 | 51.69±2.21 | 81.48±1.28 |
| | Ensemble | 73.60±4.63 | 72.60±3.45 | 73.58±2.25 | 70.29±2.26 | 90.45±1.75 | 49.60±4.26 | 53.35±2.59 | 82.52±1.24 |
| | G-MIMO | 72.70±2.53 | 73.41±4.37 | 76.16±2.47 | 72.16±3.16 | 90.15±1.73 | 50.93±3.45 | 54.05±4.05 | 82.36±1.53 |
| | GMV | **75.50±3.67** | **74.67±5.84** | **76.96±2.33** | **76.86±2.15** | **91.40±2.26** | **51.53±2.58** | **54.15±3.15** | **83.92±1.73** |
| | Vanilla | 71.70±3.10 | 64.70±6.42 | 78.47±2.41 | 78.97±1.72 | 90.10±1.77 | 48.67±3.75 | 53.89±2.15 | 80.48±1.37 |
| | DropEdge | 71.70±4.03 | 68.29±4.01 | 76.45±2.76 | 75.33±2.02 | 89.90±2.17 | 50.00±4.38 | 54.19±2.23 | 79.78±1.65 |
| | DropNode | 74.00±4.63 | 72.51±2.53 | 78.98±1.86 | 78.77±1.92 | 90.55±1.92 | 51.00±3.00 | 55.23±2.34 | 80.16±1.71 |
| | Subgraph | 73.20±3.25 | 72.24±5.76 | 77.57±2.71 | 77.32±1.71 | 88.50±2.97 | 49.07±3.84 | 53.37±2.61 | 80.66±1.75 |
| | M-Mixup | 73.10±4.21 | 71.97±3.75 | 78.52±2.05 | 81.03±0.88 | 82.25±3.87 | 49.80±3.90 | 51.49±2.01 | 80.18±1.31 |
| GIN | G-Mixup | 72.40±5.64 | 64.69±3.60 | 78.20±1.58 | 79.75±2.70 | 90.20±2.84 | 49.93±2.82 | 54.33±1.99 | 80.18±1.62 |
| | Submix | 72.50±4.94 | 69.81±4.57 | 82.90±2.45 | 81.04±1.57 | 90.20±1.95 | 49.80±4.22 | 54.59±3.29 | 82.60±1.73 |
| | S-Mixup | 72.80±3.82 | 67.57±3.50 | 69.03±1.61 | 69.57±2.56 | 87.00±4.25 | 48.53±3.38 | 52.75±2.53 | 79.50±1.25 |
| | Ensemble | 74.00±3.10 | 73.50±3.04 | 80.34±2.56 | 80.15±1.83 | **92.70±1.87** | 49.80±2.91 | 55.19±2.58 | 81.58±1.55 |
| | G-MIMO | 73.40±2.23 | 73.70±2.65 | 80.83±1.83 | 81.02±2.49 | 91.50±1.88 | 50.40±4.78 | 55.03±3.01 | 81.24±1.50 |
| | GMV | **74.20±3.37** | **74.40±3.95** | **82.38±2.15** | **82.53±1.95** | 92.50±1.30 | **52.27±3.67** | **55.35±2.41** | **83.02±1.47** |

Table 1: Comparison between GMV and other baselines are conducted on TUDataset benchmark.

### 3.3.3 Multi-view Learning

During training, we employ a three-layer multilayer perceptron (MLP) as a predictor to simultaneously classify diverse views. Unlike traditional ensemble methods, we simply double the output dimension of the predictor, transforming it into a dual-output predictor that generates two outputs. It can guide the shared backbone to facilitate the cost-effective realization of two sub-networks:

$$\hat{\mathbf{y}}_i^1, \hat{\mathbf{y}}_i^2 = \text{Predictor}(\mathbf{p}_i), \tag{7}$$

where $i \in \{\text{src}, \text{trg}, \text{mix}\}$. Moreover, to optimize GNN with these diverse views, we propose the mixed-view loss $\ell_{\text{mix}}$ and the multi-view loss $\ell_{\text{view}}$:

$$\ell_{\text{mix}} = w_{\text{src}}\text{CE}(\hat{\mathbf{y}}_{\text{mix}}^1, \mathbf{y}_{\text{src}}) + w_{\text{trg}}\text{CE}(\hat{\mathbf{y}}_{\text{mix}}^2, \mathbf{y}_{\text{trg}}), \tag{8}$$

$$\ell_{\text{view}} = \text{CE}(\hat{\mathbf{y}}_{\text{src}}^1, \mathbf{y}_{\text{src}}) + \text{CE}(\hat{\mathbf{y}}_{\text{trg}}^2, \mathbf{y}_{\text{trg}}), \tag{9}$$

$w_{\text{src}}$ and $w_{\text{trg}}$ are considered as confidence of labels of two graphs, calculated in Equation (2). The mixed-view loss $\ell_{\text{mix}}$ helps GNN inferring partial labels of $\mathcal{G}_{\text{src}}$ and $\mathcal{G}_{\text{trg}}$, playing a role of regularization, while the multi-view loss $\ell_{\text{view}}$ directly boosts the capacity of diverse view representations of GNN. These two losses collectively improve the diversity of sub-networks integrated into GNN, enhancing the generalization and robustness:

$$\ell = \ell_{\text{mix}} + \alpha\ell_{\text{view}} + R(\theta), \tag{10}$$

where $\ell$ is the final loss, $\alpha$ is the hyper parameter and $R(\theta)$ denotes the regularization item, e.g., $l_2$ norm. The detail of multi-view learning process is in the Algorithm 3 of Appendix 6.2.

### 3.4 Inference

During inference, GMV processes unseen input $\mathcal{G}_{\text{test}}$ via a standard forward pass. The primary distinction of GMV from standard GNNs lies in its dual prediction heads. Unlike the training phase, subgraph processing and multi-view decomposition are not required during inference. The final prediction is obtained by averaging the outputs of the dual prediction heads. This approach effectively acts as an efficient ensemble within a single model, leveraging the benefits of multi-view learning:

$$f_\theta\left(\hat{\mathbf{y}}_{\text{test}} \mid \mathcal{G}_{\text{test}}\right) = \frac{1}{2}\sum_{m=1}^{2} f_\theta\left(\hat{\mathbf{y}}^{(m)} \mid \mathcal{G}_{\text{test}}\right). \tag{11}$$

# 4 Experiments

**Baselines.**

GCN [19], GIN [21] are utilized as GNN backbones, and GraphGPS [23] is selected as the GT backbone. We evaluate our effectiveness of GMV compared with graph augmentation methods, such as DropEdge [6], DropNode [30] Subgraph [31], M-Mixup [48] , G-Mixup [49] and SubMix [33].

For ensemble learning [9], we consider an classic ensemble and G-MIMO [37]. For fair comparison, we only consider ensemble of two networks/sub-networks.

| | Method | HIV | BBBP | BACE | PPA |
|---|---|---|---|---|---|
| | #graphs | 41127 | 2039 | 1513 | 158100 |
| | #classes | 3 | 2 | 2 | 2 |
| | #avg nodes | 25.5 | 24.1 | 34.1 | 243.4 |
| | #avg edges | 54.9 | 26.0 | 36.9 | 2266.1 |
| GCN | Vanilla | 75.38±0.21 | 65.74±0.17 | 77.74±0.23 | 68.33 ±0.33 |
| | Submix | 75.63±0.17 | 65.90±0.54 | 78.00±0.32 | 68.97 ±0.39 |
| | G-MIMO | 75.97±0.18 | 65.87±0.40 | 78.23±0.35 | 70.02 ±0.32 |
| | GMV | **76.16±0.15** | **66.18±0.10** | **78.51±0.32** | **70.21 ±0.21** |
| GIN | Vanilla | 76.01±0.11 | 66.34±0.32 | 78.42±0.42 | 69.00 ±0.18 |
| | Submix | 77.00±0.46 | 67.67±0.29 | 78.93±0.43 | 70.43 ±0.23 |
| | G-MIMO | 77.43±0.23 | 68.38±0.43 | 78.89±0.13 | 70.08 ±0.18 |
| | GMV | **78.23±0.43** | **68.56±0.31** | **79.43±0.28** | **71.56 ±0.17** |
| GraphGPS | Vanilla | 77.53±0.80 | 67.84±1.65 | 80.54±0.87 | 80.15±0.12 |
| | Submix | 78.47±0.94 | 68.38±1.21 | 81.21±0.25 | 80.60±0.33 |
| | G-MIMO | 78.65±1.04 | 68.78±0.86 | 82.07±2.59 | 80.88±0.21 |
| | GMV | **80.23±1.02** | **70.32±0.94** | **83.99±0.17** | **81.21±0.32** |

Table 2: Comparison between GMV and other baselines are conducted on four OGB benchmark datasets.

**Experiment Details.** For each method, we conduct 10-fold cross-validation experiments on each dataset from TUDataset Benchmark, calculating the mean accuracy and standard deviation to derive results. Following S-Mixup [7], the datasets are split into training, validation and test sets. Specifically, 80% for training, 10% for validation, and 10% for testing. For the datasets from OGB Graph Banchmark [50], we adopt the public train/validation/test splits, and report the results of the test set. We conduct each experiment three times and utilize area under curve (AUC) as measurement on these OGB graph datasets. All experiments are conducted on NVIDIA 3090TI GPUs.

**Datasets.** We consider different sizes and numbers of graphs to evaluate the performance of our proposed method. Table 1 and Table 2 outlines the specifics of eight real-world datasets from the TUDatasets benchmark [51] and three datasets from open graph benchmark (OGB) [52].

## 4.1 Overall Comparison

Table 1 and Table 2 presents the results of GNNs with GMV alongside other baselines across eight benchmark datasets from TUDataset and four benchmark datasets from OGB. By simultaneously incorporating multi-view learning from the perspectives of model, data, and optimization, GMV significantly improves the average accuracy of both GCN and GIN on the TUDataset benchmark datasets. Unlike other graph augmentation and ensemble methods, which typically expand the "view" from a single perspective, GMV offers a unified and efficient approach.

To evaluate the effectiveness of GMV on large-scale graph classification tasks, we use the widely adopted GraphGPS [23] as the backbone for experiments on OGB datasets and TUDataset. As shown in Table 6a and Table 2, GMV achieves the best performance across all tested datasets. This approach has established state-of-the-art results, further highlighting GMV's superiority over traditional methods. In Appendix 6.3, Table 6, we also conduct experiments on state-of-arts GNNs.

## 4.2 Generalization and Robustness

**Limited Labels for GMV.** Following NoisyGL [53], we conduct the comparison study on limited and noisy labeled graph data to demonstrate robustness and generalization of GMV. We adopt 75%, 50%, 25% and 10% training label ratios to verify the generalization of GMV. As shown in Fig 3(a), GMV consistently outperforms other methods with different label ratios, thereby achieving great generalization.

**Noisy Labels for GMV.** To simulate label noise, we randomly corrupt 10%, 20% and 40% training labels on IMDBB and PROTEINS datasets, while keeping validation and testing datasets unchanged.

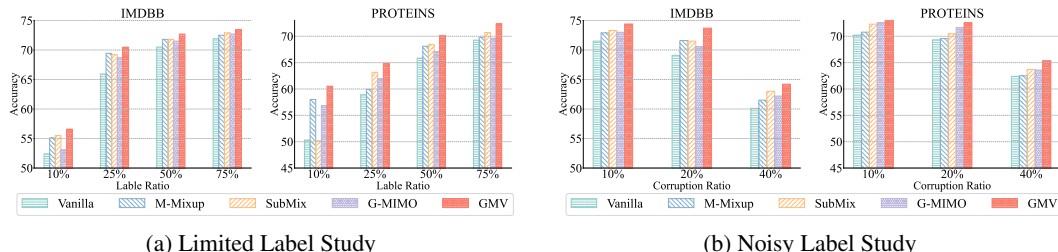

(a) Limited Label Study        (b) Noisy Label Study

Figure 3: Comparison study between GMV and other methods for different ratio of label/varying levels of label corruption on IMDBB and PROTEINS for GCN.

|  | Vanilla | Random | PPR | BFS | DPP | DPP w. BFS | ST-PPR |
|---|---|---|---|---|---|---|---|
| IMDBB | 72.30±2.84 | 72.70±5.16 | 72.60±2.37 | 73.00±4.36 | 72.60±3.69 | 73.20±4.26 | **74.10±3.01** |
| PROTEINS | 72.15±3.75 | 72.60±2.37 | 73.05±3.70 | 72.73±1.90 | 72.84±1.77 | 72.63±2.61 | **74.27±1.61** |

Table 3: Comparisons among different subgraph sampling methods for GCN.

As shown in Figure 3(b), GMV achieves better results under different noisy condition, which evaluate the robustness of it.

### 4.3 Ablation Study

**Comparison of View Generation Methods.** We first compare the effectiveness of our proposed ST-PPR with other subgraph sampling methods. Vanilla indicates the GCN without graph augmentations. As shown in Table 3, our subgraph sampling method achieves the best performance among them because it considers both structure and semantic information. Moreover, we investigate the effectiveness of ST-PPR, SubMix and ST-SubMix. As depicted in Table 4(a), ST-SubMix achieves higher accuracy than SubMix by considering the property of the structure. In Appendix 6.4, Table 7b, we also compare different graph augmentation methods for G-MIMO to generate richer training samples. These methods yield lower accuracy than GMV, thereby verifying the effectiveness of GMV.

**Ablation of MVG and MVD.** We examine the efficacy of mixed-view generation (MVG) and multi-view decomposition (MVD) for GMV (GCN) on the IMDBB and PROTEINS datasets. The results, reported in Table 4(b), show that both MVG and MVD play a crucial role in enhancing performance. Combining these two achieves the best performance, which implies that expanding views from both data and model perspectives simultaneously can help the model learn better multi-view representations. More details can be found in the When only "MVG" is applied, GMV enhances GNN performance from a data perspective, playing a same role of ST-SubMix. In contrast, with only "MVD" GMV boosts GNNs from a model perspective. With consistent graph pair inputs, GMV modifies the GNN structure in a manner the same as G-MIMO [37]. Unlike simply increasing the size of the prediction head [54], this approach leverages distinct graphs to activate different sub-networks within the GNN, achieving a simple ensemble. These two methods respectively improve of GCN, as shown in Table 4b.

**Ablation of Mixed-view/Multi-view Loss.** Additionally, we conduct an ablation study to verify the impact of mixed-view loss and multi-view loss in the GMV framework on the IMDBB dataset. As shown in Table 4(c), these two losses collectively enhance the accuracy of the GNN. When we only adopt each of these losses, GMV achieves lower accuracy than when both are considered. Therefore, both losses are necessary to encourage sub-networks to learn from mixed and multi-views, thereby enhancing the multi-view learning ability from an optimization perspective.

### 4.4 Efficiency Study

During inference, GMV requires only a single forward pass of standard GNNs with an additional prediction head. Consequently, GMV's time complexity is nearly identical to that of standard GNNs, as illustrated in Fig 1, where GMV demonstrates the optimal balance between accuracy and computational overhead. As for training, the mixed-view generation process can be preprocessed only once to obtain sampled nodes for each graph, therefore significantly accelerating the training

| Methods | IMDBB | PROTEINS |
|---|---|---|
| Vanilla | 72.30±4.34 | 72.15±3.75 |
| SubMix | 73.80±3.57 | 73.50±5.38 |
| ST-PPR | **74.10±3.01** | 72.87±4.09 |
| ST-SubMix | **74.10±3.66** | **74.40±5.98** |

(a) Comparison of VG

| /w. MVG | /w. MVD | IMDBB | PROTEINS |
|---|---|---|---|
| | | 72.30±4.34 | 72.15±3.75 |
| ✓ | | 74.10±3.66 | 74.40±5.98 |
| | ✓ | 72.70±2.53 | 73.41±4.37 |
| ✓ | ✓ | **75.50±3.67** | **74.67±5.84** |

(b) Ablation of Components

| /w. $\ell_{mix}$ | /w. $\ell_{view}$ | IMDBB | PROTEINS |
|---|---|---|---|
| | | 72.30±4.34 | 72.15±3.75 |
| ✓ | | 74.55±2.32 | 74.60±2.38 |
| | ✓ | 74.55±3.18 | 73.87±3.95 |
| ✓ | ✓ | **75.50±3.67** | **74.67±5.84** |

(c) Ablation of Losses

Table 4: Results of ablation studies. (a) Comparison of different view generation methods (VG) including our proposed ST-PPR and ST-SubMix. (b) Ablation of two components of our proposed GMV. (c) Ablation of our proposed mixed-view loss ($\ell_{mix}$) and multi-view loss ($\ell_{view}$).

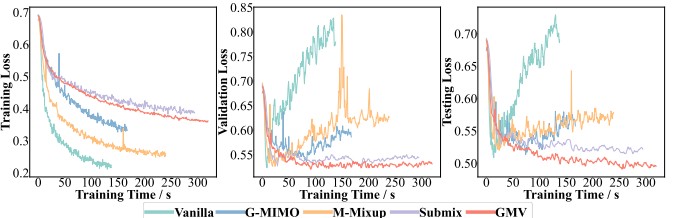

Figure 4: Training Time v.s. Training/Validation/Testing Loss on IMDBB.

| | $D_{\text{Disagree}}$ | $D_{\text{KL}}$ | Accuracy |
|---|---|---|---|
| Vanilla | 0 | 0 | 70.27 |
| Submix | 0 | 0 | 72.91 |
| Ensemble | 10.02 | 1.56 | 70.29 |
| G-MIMO | 12.14 | 1.63 | 72.16 |
| GMV | 13.40 | 5.41 | **76.86** |

Table 5: Comparison of prediction on NCI109 between dual subnetworks of GMV and baselines within GCN.

process. Specifically, given $\mathcal{G}_{\text{src}}$ and $\mathcal{G}_{\text{trg}}$, the time complexity of mixed-view generation process is $O(|\mathcal{V}_{\text{src}}| + |\mathcal{V}_{\text{trg}}|)$. We monitor the evolution of training and validation loss over time in Fig 4. While the vanilla GCN converges fastest, it suffers from significant overfitting. In contrast, graph augmentation techniques like M-Mixup and Submix, along with the ensemble method G-MIMO, help mitigate overfitting to some extent. our GMV framework inherently functions as a more powerful regularizer compared to these standard methods. This is evidenced by GMV achieving a lower validation loss and, consequently, better generalization to the test set.

## 4.5 Quantitative Study of Diversity.

We evaluate the diversity of predictions made by GCN within GMV and other baseline methods on the NCI109 dataset. We employ disagreement [11]($D_{\text{Disagree}}$) and average Kullback-Leibler divergence [13] ($D_{\text{KL}}$) as diversity metrics. Suppose $f_1$ and $f_2$ are two (sub-)networks. $D_{\text{Disagree}}$ is computed as $\sum_{\mathcal{G} \in \mathbb{G}} \mathbb{1}(f_1(\mathcal{G}) \neq f_2(\mathcal{G}))$, where $\mathbb{1}(\cdot)$ equals 1 only if $f_1(\mathcal{G}) \neq f_2(\mathcal{G})$. $D_{\text{KL}}$, is calculated as $\frac{1}{2}(\text{KL}(\hat{y}_1||\hat{y}_2) + \text{KL}(\hat{y}_2||\hat{y}_1)) = \frac{1}{2}(\mathbb{E}_{\hat{y}_2}(\log \hat{y}_2 - \log \hat{y}_1) + \mathbb{E}_{\hat{y}_1} \log(\hat{y}_1 - \log \hat{y}_2))$. As shown in Table 5, GMV achieves higher $D_{\text{Disagree}}$, $D_{\text{KL}}$ and accuracy, indicating an enhanced capacity to represent diverse views for better generalization.

## 5 Conclusion

We have introduced GMV, an unified and efficient framework that significantly enhances the robustness and generalization capabilities of GNNs/GTs in graph classification. During training, GMV encourages GNNs/GTs to explore diverse views by integrating data, model, and optimization perspectives through a mixed view generation and multi-view decomposition and learning pipeline. During inference, GMV appends an additional prediction head to standard GNNs/GTs, enabling superior performance in a single forward pass with ensemble-like behavior. Our extensive experiments across various datasets demonstrate that GMV consistently outperforms existing augmentation and ensemble techniques, establishing it as a highly effective and promising method to improve the performance and generalization of GNNs/GTs.

## Acknowledgments and Disclosure of Funding

This work is supported by National Natural Science Foundation of China (NSFC 62176059, 62576103). The computations in this research were performed using the CFFF platform of Fudan University.

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

# 6 Appendix

## 6.1 Proof of ST-PPR Algorithm

**Theorem**: Let a subgraph sampling strategy $S$ generates a subgraph. Define structural preservation score $\rho(\mathcal{G}_s)$ as the graph kernel similarity between $\mathcal{G}_s$ and the original graph $\mathcal{G}$: $\rho(\mathcal{G}_s) = \frac{\langle \phi(\mathcal{G}), \phi(\mathcal{G}_s) \rangle}{\|\phi(\mathcal{G})\| \cdot \|\phi(\mathcal{G}_s)\|}$, where $\phi(\cdot)$ is a graph kernel mapping function. For any graph $\mathcal{G}$, there exist constants $\epsilon_1, \epsilon_2, \epsilon_3 > 0$ such that: $\rho(\mathcal{G}_{\text{integ}}) \geq \max\{\rho(\mathcal{G}_{\text{PPR}}), \rho(\mathcal{G}_{\text{BFS}}), \rho(\mathcal{G}_{\text{DFS}})\} + \epsilon_{\text{integ}}$, where $\epsilon_{\text{integ}}$ represents the gain from integration. This ensures comprehensive feature extraction including global topology , hierarchical transitions and local communities, which boosts the performance of GNNs.

**Proof**: PPR selects high-centrality nodes via its stationary distribution $\pi$. For any node $u$, its PageRank value satisfies: $\pi(u) = \alpha \sum_{v \in \mathcal{V}} \pi(v) \frac{A_{vu}}{d(v)} + (1 - \alpha) q(u)$, where $d(v)$ is the degree of $v$, and $q(u)$ is the initial distribution. High-$\pi(u)$ nodes form the backbone of $\mathcal{G}$, ensuring $\rho(\mathcal{G}_{\text{PPR}}) \geq \epsilon_1$. For any node $u$, its local clustering coefficient $C(u)$ in BFS subgraph satisfies: $C_{\text{BFS}}(u) \geq C_G(u) - \delta_1$, where $\delta_1$ bounds sampling error. Thus, $\rho(\mathcal{G}_{\text{BFS}}) \geq \epsilon_2$. DFS retains long-range dependencies. Let $D$ be the diameter of $G$. The diameter of the DFS subgraph $D_{\text{DFS}}$ satisfies: $D_{\text{DFS}} \geq D - \delta_2$, where $\delta_2$ bounds path truncation error. Hence, $\rho(\mathcal{G}_{\text{DFS}}) \geq \epsilon_3$. The joint structural representation is: $\phi(\mathcal{G}_{\text{integ}}) = \phi(\mathcal{G}_{\text{PPR}}) \oplus \phi(\mathcal{G}_{\text{BFS}}) \oplus \phi(\mathcal{G}_{\text{DFS}})$, where $\oplus$ denotes node concatenation. By linearity of kernel functions: $\rho(\mathcal{G}_{\text{integ}}) \geq \max\{\rho(\mathcal{G}_{\text{PPR}}), \rho(\mathcal{G}_{\text{BFS}}), \rho(\mathcal{G}_{\text{DFS}})\}$ . When structural information from three strategies is non-overlapping, $\epsilon_{\text{integ}} > 0$.

## 6.2 Algorithm of Multi-view Decomposition and Learning

---

**Algorithm 3** Multi-view Decomposition and Learning

---

**Input**: Graph dataset $\mathbb{G} = \{(\mathcal{G}_t, y_t)\}_{t=1}^n$, the graph model $f_{\text{GMV}}$, loss weight $\alpha$
**Output**: Trained graph model $f_{\text{GMV}}$

1: **while** not convergence **do**
2:     **for** src = 1 : n **do**
3:         $\mathcal{G}_{\text{trg}}, y_{\text{trg}} \leftarrow$ randomly sample a graph from $\mathbb{G}/\{\mathcal{G}_{\text{src}}\}$
4:         $\mathcal{G}_{\text{mix}}, \mathbf{E}_{\text{src}}, \mathbf{E}_{\text{trg}}, w_{\text{src}}, w_{\text{trg}} \leftarrow$ employ ST-SubMix between graph $\mathcal{G}_{\text{src}}$ and $\mathcal{G}_{\text{trg}}$
                                                        ▷ ST-SubMix 2
5:         $\hat{y}_{\text{src}}^1, \hat{y}_{\text{trg}}^2, \hat{y}_{\text{mix}}^1, \hat{y}_{\text{mix}}^2 \leftarrow f_{\text{GMV}}(\mathcal{G}_{\text{mix}}, \mathbf{E}_{\text{src}}, \mathbf{E}_{\text{trg}})$
6:         $\ell_{\text{mix}} \leftarrow w_{\text{src}}\text{CE}(\hat{y}_{\text{mix}}^1, y_{\text{src}}) + w_{\text{trg}}\text{CE}(\hat{y}_{\text{mix}}^2, y_{\text{trg}})$
7:         $\ell_{\text{view}} \leftarrow \text{CE}(\hat{y}_{\text{src}}^1, y_{\text{src}}) + \text{CE}(\hat{y}_{\text{trg}}^2, y_{\text{trg}})$
8:         $\ell \leftarrow \ell_{\text{mix}} + \alpha \ell_{\text{view}} + \text{R}(\theta)$
9:         Update parameters of the model $f_{\text{GMV}}$
10:     **end for**
11: **end while**

---

In Algorithm 3, we generate a mixed-view and feed it into the GNNs. We then perform multi-view decomposition and predict the labels for each of the decomposed diverse views. To activate the dual sub-networks in the GNNs, we minimize both the mixing loss and the multi-view loss, thereby enhancing the multi-view representation of the GNNs.

## 6.3 Comparison Study

To validate the efficacy of our proposed GMV method, we conduct a series of comparison studies. As shown in Table 6a, within the GraphGPS framework, GMV outperforms baseline methods, including Vanilla and G-MIMO, on both the IMDBB and PROTEINS datasets. Furthermore, to examine its generality, we apply GMV to several mainstream GNN backbones. The results in Table 6b indicate that GMV can serve as a plug-and-play module, consistently improving the performance of GatedGCN, GINE, and NSA [55] across multiple molecular graph datasets, thereby demonstrating its broad applicability and effectiveness.

| Method | IMDBB | PROTEINS |
|---|---|---|
| Vanilla | 74.50 ±4.53 | 74.76 ± 3.24 |
| Submix | 75.34 ±3.68 | 75.21 ±1.42 |
| G-MIMO | 75.68 ±4.34 | 75.08 ±3.32 |
| **GMV** | **76.70 ±3.22** | **75.78 ±4.13** |

(a) Comparison on the GraphGPS framework.

| | HIV | BBBP | BACE |
|---|---|---|---|
| GatedGCN | 76.39 | 67.05 | 78.75 |
| **/w. GMV** | **77.04** | **69.43** | **79.86** |
| GINE | 76.45 | 67.56 | 77.91 |
| **/w. GMV** | **77.76** | **70.30** | **78.82** |
| NSA [55] | - | 84.0 | 72.0 |
| **/w. GMV** | - | **85.50** | **74.1** |

(b) Comparison on different backbones.

Table 6: Comparison studies evaluating the effectiveness and generality of our proposed GMV method. (a) Performance comparison against other methods on the GraphGPS framework. (b) Generality study by integrating GMV with different GNN backbones.

## 6.4 Ablation Study

**Ablation of Mixup.** From a data perspective, we compare various mixup strategies for mixed-view generation. As shown in Table 7a, GMV consistently achieves higher accuracy than other mixup methods, demonstrating its effectiveness. The full GMV enhances the ability of multi-view representation from data, model and optimization perspectives, including mixed-view generation, multi-view decomposition and multi-view learning. M-Mixup linearly interpolates graph representations to create mixed-views, making it difficult to apply multi-view decomposition and learning. S-Mixup uses a trained graph matching transformer to map the source graph to the target graph, which distorts the information of the source graph and hinders multi-view decomposition and learning. "GMV w. M-Mixup" and "GMV w. S-Mixup" only employ mixing loss to optimize dual sub-networks within GNNs. In contrast, SubMix and ST-SubMix generate mixed-views by connecting subgraphs, preserving subgraph view information, and enabling them to consider three perspectives concurrently. "GMV w. SubMix" and "GMV w. ST-SubMix" simultaneously consider mixed-view generation, multi-view decomposition and learning to enhance the performance of GNNs. Consequently, they outperform GMV with other mixup methods. SubMix focuses on semantic information, while ST-SubMix considers both structural and semantic information to create structure enhanced subgraph views, thus achieving state-of-the-art performance and generalization for GNNs.

**Further Comparation with MIMO.** In this section, we perform additional experiments on G-MIMO with various augmentations and observe that graph augmentations combined with ensemble learning enhance GNN performance. As shown in Table 7b, integrating G-MIMO with drop-based augmentations improves GCN accuracy on IMDBB. Different augmentations create diverse views

| Method | GCN | GIN |
|---|---|---|
| Vanilla | 72.30±2.84 | 71.70±3.10 |
| M-Mixup | 73.70±4.12 | 73.10±4.21 |
| S-Mixup | 72.50±2.20 | 72.80±3.82 |
| SubMix | 73.80±3.57 | 72.50±4.94 |
| ST-SubMix | 74.00±3.66 | 74.50±3.32 |
| **GMV /w. M-Mixup** | 72.40±2.33 | 74.10±3.96 |
| **GMV /w. S-Mixup** | 73.10±4.12 | 74.00±4.15 |
| **GMV /w. SubMix** | 75.00±4.28 | 74.10±3.32 |
| **GMV /w. ST-SubMix** | 75.50±3.67 | 74.20±3.37 |

(a) Ablation on mixup methods.

| Method | Accuracy |
|---|---|
| Vanilla GCN | 72.30±2.84 |
| G-MIMO | 72.70±2.53 |
| G-MIMO w. DropNode | 73.50±4.30 |
| G-MIMO w. DropEdge | 72.50±2.84 |
| G-MIMO w. Subgraph (R) | 73.40±4.15 |
| G-MIMO w. Subgraph (PPR) | 74.10±4.72 |
| G-MIMO w. Subgraph (ST-PPR) | 74.40±4.33 |
| **GMV** | **75.50±3.67** |

(b) Ablation on augmentation types.

Table 7: Ablation studies on the IMDB-BINARY dataset. All results are based on the GCN backbone. (a) Comparison of different mixup strategies. Our full model, "GMV /w. ST-SubMix", achieves the best performance. (b) Comparison of GMV against various augmentation techniques used in G-MIMO.

that boost performance of G-MIMO. The utilization of mixed-view generation provides richer view information, activating sub-networks in GNNs for enhanced representations. Additionally, GMV combines mixed-view generation and multi-view decomposition, enabling effective multi-view learning.

**Performance vs. number of sub-networks.** To assess framework scalability and efficiency, we compare GMV against G-MIMO by varying the number of sub-networks. The results in Table 8 are striking. GMV not only consistently outperforms G-MIMO, but its efficiency is such that using only two sub-networks (75.50%) already surpasses a 10-sub-network G-MIMO (75.30%). This significant performance gain stems from GMV's integrated design, which fosters more diverse and complementary predictions among the generated views, leading to stronger generalization. All results are based on a rigorous and fair comparison protocol.

| Sub-nets | G-MIMO | GMV |
|----------|--------|-----|
| 2 | 72.70 | **75.50** |
| 4 | 74.40 | **75.90** |
| 6 | 74.52 | **76.10** |
| 8 | 74.73 | **76.12** |
| 10 | 75.30 | **76.43** |

Table 8: Performance vs. number of sub-networks on IMDB-B. GMV shows superior efficiency.

## 6.5 Hyperparameter Analysis

We conducted a sensitivity analysis on key hyperparameters: the feature augmentation ratio ($p$), the structure augmentation ratio ($q$), and the loss weight ($\alpha$). As shown in Table 9, the results on the BACE dataset demonstrate the robustness of our model. Performance remains stable across a wide range of values for each hyperparameter, obviating the need for exhaustive or fragile tuning to achieve strong results. Notably, the optimal values fall within conventional ranges guided by prior work, reinforcing the model's stability and ease of adoption. To ensure full reproducibility, our complete source code and detailed settings will be made publicly available.

| Augmentation Ratio ($p$) | | Structure Ratio ($q$) | | Loss Weight ($\alpha$) | |
|------|----------|------|----------|------|----------|
| Value | Accuracy | Value | Accuracy | Value | Accuracy |
| 0.2 | 78.82 | 0.2 | 78.93 | 0.5 | 78.57 |
| **0.4** | **79.43** | 0.4 | 78.98 | 1.0 | 78.98 |
| 0.5 | 79.32 | 0.5 | 79.18 | **2.0** | **79.43** |
| 0.6 | 79.02 | **0.6** | **79.43** | | |
| 0.8 | 78.72 | 0.8 | 78.34 | | |

Table 9: Hyperparameter sensitivity analysis on the BACE dataset with a GIN backbone. The model exhibits robustness, with stable performance across a wide range of values. The best-performing setting for each hyperparameter is highlighted in **bold**.

## 6.6 Efficiency Study

We provide a transparent analysis of our method's computational cost, examining both the one-time preprocessing overhead and the online training efficiency.

**One-Time Preprocessing Cost.** Our method requires a one-time, offline preprocessing step to generate and cache views. As shown in Table 10, this cost is negligible. On the PROTEINS dataset, it amounts to less than five minutes, which is merely 0.4% of the total training time. This efficiency scales to the larger COLLAB dataset, where the 2-hour preprocessing cost is only 1.1% of the 180-hour training duration. This fixed cost is comparable to other advanced augmentation methods and is incurred only once, making it a highly practical investment.

| Dataset | Graph Count | Preprocessing (Hours) | Total Training (Hours) |
|---------|-------------|------------------------|------------------------|
| PROTEINS | 1,113 | ∼0.08 | 20 |
| COLLAB | 5,000 | ∼2 | 180 |

Table 10: Offline preprocessing cost analysis. The one-time cost is minimal compared to the total training time (10-fold CV) on an NVIDIA 3090Ti GPU.

**Online Training Overhead vs. Performance Gain.** The online training phase is lightweight. Since all views are pre-computed and cached, the only overhead stems from view lookups and the forward passes for the sub-networks. Table 11 quantifies the trade-off between this training overhead and the resulting accuracy improvement over a GCN baseline. The results clearly show that for a manageable training overhead of +110-125%, our method delivers a substantial and consistent accuracy gain of approximately +9% across all datasets. This demonstrates a highly favorable and predictable return on computational investment, confirming the practical value of our approach.

| Dataset | Num. Graphs | Avg. Edges | Training Overhead | Accuracy Gain |
|---|---|---|---|---|
| NCI1 | 4,110 | 32.3 | +113% | +9.4% |
| PROTEINS | 1,113 | 72.8 | +120% | +8.8% |
| COLLAB | 5,000 | 2,457.2 | +125% | +9.0% |

Table 11: Training time overhead vs. accuracy gain over a GCN baseline. A manageable increase in training time yields a significant and consistent performance improvement.

## 6.7 Multi-view Study

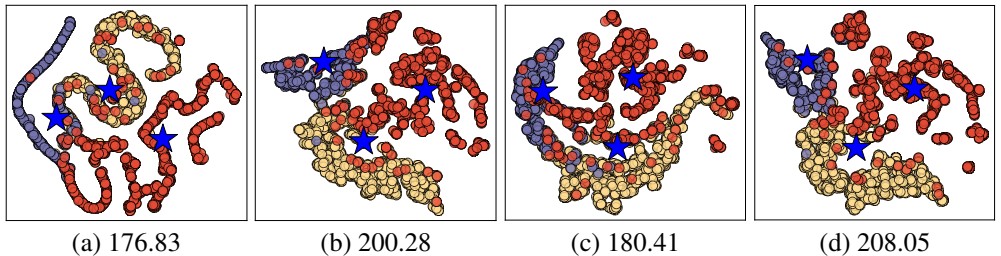

(a) 176.83  (b) 200.28  (c) 180.41  (d) 208.05

Figure 5: T-SNE among prediction outputs of vanilla GIN and GMV. (a) vanilla GIN; (b) and (c) two sub-networks within GMV; (d) GMV. The blue pentagrams denote three class center, and the digit is the distance among three class centers.

**Visualization of Multi-view Representation.** We employ both qualitative and quantitative methods to assess the diversity of predictions, thereby investigating the multi-view learning capacity of GMV. In Fig 5 presents the t-SNE for the vanilla GIN, two sub-networks of GIN within GMV and GMV itself, as applied to the COLLAB dataset. Different colored circles denote three classes in COLLAB, while pentagrams mark the class centers of three classes. We observe a significant difference between the two predictions, affirming the diversity of sub-networks. Moreover, the digit represents the sum of normalized $l_2$ distances among three centers. GMV achieves the largest distance among classes, which also validates the benefits of multi-view learning.

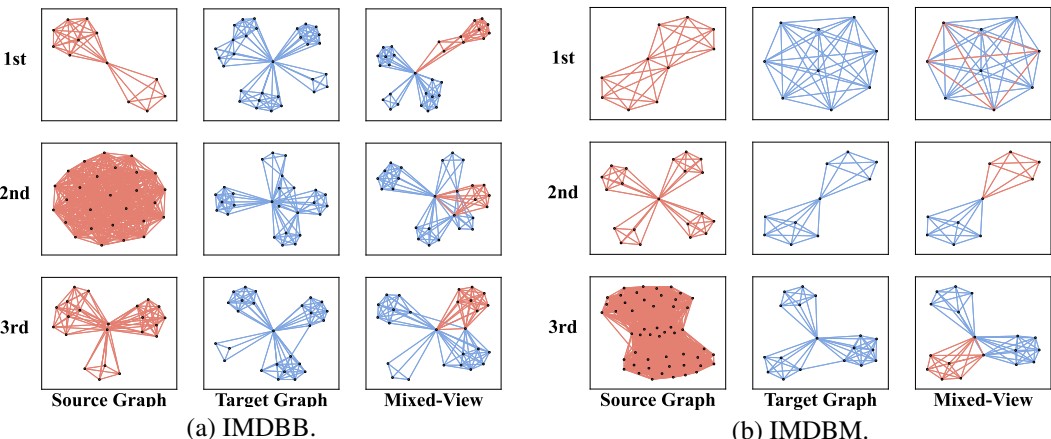

(a) IMDBB.  (b) IMDBM.

Figure 6: Visualization of mixed-views on IMDBB and IMDBM.

**Visualization of Mixed-view.** We utilize networkx to visualize some mixed-views in Fig 6. Each row denotes the source graph, target graph and generated mixed-view. ST-SubMix consider both structure and semantic information, so it generates the subgraph views preserving the original topology structure and semantic key nodes. ST-SubMix generates diverse mixed-views for GMV to enhance multi-view representation of capacity of GNNs.

## 6.8   Discussion

The framework naturally extends to other crucial tasks, such as node classification and link prediction. This is achieved by leveraging the powerful paradigm of task reformulation, where local tasks are converted into graph-level problems, a strategy validated by recent work. This requires minimal architectural changes: For Node Classification: The task can be reframed as classifying a node's contextual subgraph. GMV is then applied directly to this subgraph to predict the central node's label, thereby benefiting from a robust, multi-view representation of its neighborhood. For Link Prediction: Similarly, this becomes a binary classification problem on the subgraph enclosing a pair of nodes. GMV's ability to capture diverse and subtle topological patterns makes it ideally suited for predicting the existence of a link between them. Furthermore, the core principles of GMV are adaptable to more complex domains, such as dynamic graphs (by applying the framework to temporal snapshots) and heterogeneous graphs (by acting as a modular wrapper around specialized GNN backbones).

