# OpenReview forum: "GMV: A Unified and Efficient Graph Multi-View Learning Framework"
_NeurIPS.cc/2025/Conference — NeurIPS 2025 poster_

### Official Review · Reviewer_GVQ8 · 2025-06-21

**Clarity:** 3
**Significance:** 3
**Originality:** 3
**Rating:** 4
**Confidence:** 1

**Summary:**

This paper presents ​​Graph Multi-View (GMV)​​, an innovative framework that enhances the generalization and robustness of Graph Neural Networks (GNNs) and Graph Transformers (GTs) for graph classification tasks. By integrating ​​multi-view learning​​ from data, model, and optimization perspectives, GMV activates diverse sub-networks within a single GNN/GT through a novel training pipeline involving ​​structure-enhanced subgraph mixing​​ and ​​multi-view decomposition​​. During inference, GMV efficiently leverages dual prediction heads to achieve ensemble-like performance with minimal computational overhead. Extensive experiments on benchmark datasets demonstrate that GMV outperforms existing augmentation and ensemble methods, achieving state-of-the-art results while maintaining scalability for large-scale graphs. The framework's ability to preserve structural integrity and learn diverse representations is further validated through theoretical analysis and robustness tests under noisy and limited-label scenarios.

**Questions:**

There are some typographical errors that the authors need to correct.

Can the generalization ability of the GMV model be applied to unsupervised tasks?

Among the comparison algorithms, there are relatively few models from the past three years.

The paper proposes a unified framework (GMV) combining data, model, and optimization perspectives for multi-view learning. How does the interaction between these three perspectives quantitatively contribute to performance gains compared to isolated implementations?

ST-SubMix claims to preserve structural information better than prior methods (Section 3.2.1). However, Figure 6 shows mixed-view generation alters original graphs. What metrics quantify structural preservation, and how does distortion impact performance on tasks sensitive to topology ?

The experiments focus on graph classification. Could GMV's multi-view approach improve other tasks like node classification or link prediction? Are there theoretical or empirical results supporting broader applicability?

GMV adds minimal overhead during inference (Section 3.4), but training requires dual sub-network activation. How does training time scale with graph size, and does the benefit of multi-view learning diminish for very large graphs due to computational constraints?

**Ethical Concerns:**

["NO or VERY MINOR ethics concerns only"]

**Limitations:**

The performance of the GMV model depends on the characteristics of the dataset.

**Quality:**

3

**Strengths And Weaknesses:**

Introducing multi-view learning into GNNs to enhance generalization performance is a strong and valuable innovation. The theoretical foundation is also solid.

The experimental results are relatively rich and comprehensive, and in particular, the various ablation studies effectively demonstrate the validity of the proposed model.

---

> ### Author Rebuttal · Authors · 2025-07-30
>
> We are particularly grateful for the following acknowledgments:
>
> *   **Innovation and Theoretical Soundness:** We are pleased the reviewer found our core contribution—the integration of multi-view learning into GNNs—to be a **"strong and valuable innovation"** and our theoretical foundation **"solid."**
>
> *   **Experimental Validation:** We also appreciate the reviewer's recognition of our experimental evaluation as **"rich and comprehensive"** and our ablation studies as **"effective"** in demonstrating the model's validity and robustness.
>
> ### **Q1: Adaptability to Unsupervised Learning**
>
> The transition is straightforward, requiring one key modification: replacing the supervised losses with a **multi-level contrastive objective** (e.g., InfoNCE). The model would be trained to maximize mutual information between corresponding "positive pairs" generated from an input graph ($G_{\text{mix}}$) and its augmentation ($G'_{\text{mix}}$), treating other graphs as negatives.
>
> The positive pairs would be:
>
> *   **Composite View:** The full mixed graph and its augmentation ($G_{\text{mix}}, G'_{\text{mix}}$).
> *   **Decomposed Views:** The corresponding source subgraphs ($G_{\text{src}}, G'_{\text{src}}$)
>
>     and target subgraphs ($G_{\text{tag}}, G'_{\text{tag}}$).
>
> This multi-level contrastive strategy would compel the model to learn representations that are robust to augmentation and semantically consistent across both composite and decomposed views. We agree this is a very promising direction for future work.
>
> ### **Q2: Comparison with Additional State-of-the-Art (SOTA) Methods**
>
> We thank the reviewer for this valuable suggestion to benchmark against more recent SOTA models. While our initial experiments focused on foundational methods for a clear comparison, we agree that a broader evaluation is crucial.
>
> To that end, we have conducted new experiments during the rebuttal period, integrating our GMV framework as a plug-and-play enhancement module with the powerful NSA [1] model. The results, summarized below, show that GMV provides a consistent and significant performance uplift on top of this already strong baseline, demonstrating its effectiveness as a general-purpose enhancement.
>
> | **Method**           | **BBBP**  | **BACE**  |
> | -------------------- | --------- | --------- |
> | NSA (baseline)       | 84.0%     | 72.0%     |
> | **NSA + GMV (Ours)** | **85.5%** | **74.1%** |
>
> As the table illustrates, applying our GMV framework boosts the performance of the NSA backbone from 84.0% to **85.5%** on BBBP and from 72.0% to **74.1%** on BACE. These results highlight that GMV is not only a standalone framework but also a compatible module that can enhance other strong, state-of-the-art architectures.
>
> ---
>
> [1] Buterez et al. "An end-to-end attention-based approach for learning on graphs." *Nature Communications* (2025).
>
> ### **Q3: Quantitative Analysis of Framework Synergy**
>
> We thank the reviewer for this question, which highlights the core principle of our work: the **synergistic interplay** between our framework's data, model, and optimization perspectives. Our ablation studies in Table 4 provide direct quantitative evidence for this.
>
> **1. Synergy between Data (MVG) and Model (MVD):**
> On IMDB-BINARY (Table 4b of the main text), the baseline accuracy of 72.30% improves to 74.10% with only our data-centric **Mixed-View Generation (MVG)** and to 72.70% with only our model-centric **Multi-View Decomposition (MVD)**. However, when **both are enabled**, accuracy reaches **75.50%**. This gain significantly exceeds the sum of the individual parts, providing clear quantitative evidence of synergy.
>
> **2. Synergy in Optimization:**
> This principle is reinforced by our optimization strategy (Table 4c). While using either the mixed-view loss ($\ell_{\text{mix}}$) or the decomposed-view loss ($\ell_{\text{view}}$) alone yields an accuracy of 74.55%, only their **joint application** unlocks the peak performance of **75.50%**. This confirms both objectives are indispensable and mutually reinforcing.
>
> In summary, these results provide decisive quantitative evidence that the three pillars of our framework are fundamentally synergistic, not merely additive.
>
> ### **Q4: The Role of Structural Information in Augmentation**
>
> Our goal is not to preserve an identical topology—which offers no learning benefit—but to introduce **controlled structural distortions**. This strategy retains semantically crucial patterns while creating challenging views that enhance model robustness, a claim we support both theoretically and empirically.
>
> **1. Quantifying Structural Preservation**
>
> Our method is designed to generate augmentations that are more structurally faithful than alternatives.
>
> * **Theoretical & Empirical Proof:** Our **ST-PPR sampling** is provably better at preserving key structural patterns than simpler methods (see Appendix for formal proof). Empirically, a topological analysis on the COLLAB dataset confirms this. As shown below, our augmentations (ST-SubMix) produce statistics significantly closer to the original graph than standard SubMix.
>
>   *Topological statistics on COLLAB. Our method's results are closer to the original.*
>
> | **Topological Statistic**      | **Original Graph** | **ST-SubMix (Ours)** | **Standard SubMix** |
> | ------------------------------ | ------------------ | -------------------- | ------------------- |
> | Average Clustering Coefficient | 0.6055             | **0.6322**           | 0.6877              |
> | Average Node Size              | 74                 | **80**               | 88                  |
>
> This data quantitatively demonstrates that our method introduces more controlled and realistic structural changes.
>
> **2. How Controlled Distortion Drives Performance**
>
> This controlled distortion is precisely what compels the model to learn more generalizable representations. The effectiveness of this strategy is confirmed by two key results:
>
> *   **Success on Topology-Sensitive Tasks:** Our SOTA performance on the highly topology-sensitive COLLAB dataset (Table 1) provides strong empirical proof that these structural variations are beneficial, enhancing the model's understanding of essential features rather than degrading it.
>
> *   **Mechanism: High-Diversity Views:** The reason for this success is revealed in Table 5. Our framework produces sub-networks with significantly higher prediction diversity ($D_{\text{Disagree}}$, $D_{\text{KL}}$) than all baselines. These structurally distinct yet accurate views are highly complementary. By forcing the model to reconcile them, our framework achieves superior generalization and a final accuracy that surpasses all other methods.
>
> ### **Q5: Generalizability to Other Graph-Based Tasks**
>
> We thank the reviewer for this excellent question. The reviewer is correct that the principles of our GMV framework are highly general and extend directly to other fundamental tasks, such as **node classification** and **link prediction**, by reformulating them as graph-level problems [1].
>
> This extension requires minimal architectural changes and allows the full power of our framework to be applied:
>
> *   **For Node Classification:** The task is transformed into classifying a node's contextual subgraph. Our ST-PPR sampling is ideally suited for extracting these meaningful neighborhoods, and GMV's multi-view approach would create robust representations of the node's environment for a more accurate prediction.
>
> *   **For Link Prediction:** This is framed as a binary classification problem on an enclosing subgraph containing both nodes. Our framework's ability to generate diverse structural views is particularly advantageous for learning the complex topological patterns that signal a connection.
>
> By adopting this graph-level perspective, GMV becomes a naturally applicable and powerful tool for these tasks.
>
> ---
>
> [1] Sun et al. "All in One: Multi-Task Prompting for Graph Neural Networks." *KDD* (2023).
>
> ### **Q6: Computational Efficiency and Scalability**
>
> We thank the reviewer for this important practical concern. Our GMV framework is designed for efficiency and scalability via a **one-time, offline pre-processing stage**.
>
> **1. Efficiency by Design:**
> Before training, we perform a single pass to generate and cache all views. The computational complexity of this step is **O(V)**, which is on par with existing methods like SubMix and ensures our more sophisticated view generation does not create a bottleneck. The online training phase is therefore lightweight, with the only overhead coming from fast view lookups and additional forward passes for the sub-networks.
>
> **2. Empirical Scalability and Performance Trade-off:**
> Our empirical analysis confirms this favorable trade-off. We measured the training overhead and accuracy gain against a GCN baseline:
>
> | **Dataset** | **Num. Graphs** | **Avg. Edges** | **Training Time Overhead** | **Accuracy Gain** |
> | :---------- | :-------------: | :------------: | :------------------------: | :---------------: |
> | NCI1        |      4,110      |      32.3      |           +113%            |     **+9.4%**     |
> | PROTEINS    |      1,113      |      72.8      |           +120%            |     **+8.8%**     |
> | COLLAB      |      5,000      |    2,457.2     |           +125%            |     **+9.0%**     |
>
> The data shows two clear results:
>
> 1.  **Substantial & Consistent Gain:** GMV delivers a significant and consistent accuracy improvement of **~+9%** that scales across all datasets.
> 2.  **Manageable Overhead:** This SOTA performance is achieved for a predictable and justified training overhead of ~+110-125%.
>
> In summary, our efficient O(V) pre-processing design enables a significant performance gain for a modest increase in training cost, confirming a highly favorable trade-off between computation and generalization power.

---

### Official Review · Reviewer_9EsQ · 2025-06-29

**Clarity:** 2
**Significance:** 3
**Originality:** 3
**Rating:** 4
**Confidence:** 2

**Summary:**

This paper proposes a multi-view learning framework for graph classification task. In details, different from exisitng multi-view methods which  empoly only one way for sub-graph extraction, the proposed method combined multiple approaches for sub-graph sampling. Furthermore, sub-graphs are mixed with structure enhancement for different graph view generation. Finally, a dual-stream predictor is employed for final prediction.

**Questions:**

1. The authors should highlight their contributions. Especially the design of the mixed-view generation part.
2. The authors need to carefully review the wording of the paper, including the definitions of many formulas.
3. What concerns me most is the experimental results. The authors need to give more detailed explanations about why the results are reported differently from the original paper.

**Ethical Concerns:**

["NO or VERY MINOR ethics concerns only"]

**Final Justification:**

During the rebuttal, the authors provided a clear explanation of the model’s novelty, and also addressed my primary concern regarding the importance of the lightweight design. This has improved the overall coherence of the paper, which led me to revise my score.

**Limitations:**

yes

**Quality:**

2

**Strengths And Weaknesses:**

Pros:
1. Compared to existing methods, the authors significantly reduce the model’s parameters and complexity while maintaining performance, which is highly beneficial for deployment in real-world scenarios.
2. The authors conduct extensive experiments to prove the effectiveness of the proposed method.

Cons:
1. The overall novelty of this paper does not appear to be very strong. The key contribution of this paper is based on the multi-view graph generation process. However, the authors appear to have simply combined three existing sampling methods—PPR, DFS, and BFS. From the implementation details, it is evident that these methods are sequentially applied to the main graph to generate subgraphs, which are then directly merged. I believe this approach alone is insufficient to support a novel contribution.
2. Some concepts or formulas are not clearly explained. For example, in line 167, what is "source graph", and what is the purpose of "target graph"? in line 164, the definition of $S_{PPR}$ is missing. Although it is explained in Appendix 6, it is important to have every formulas defined in the main body of the paper.
3. It seems not necessary for light weight design, given the existing models are already very small (only 30~300KB).
4. The experimental results of G-MIMO [1] reported in Table 1 are different from (lower than) the results reported in the original paper. The authors should give a further explanation.

[1] Zhu, Qipeng, et al. "G-MIMO: Empowering GNNs with Diverse Sub-Networks for Graph Classification." 2024 IEEE International Conference on Multimedia and Expo (ICME). IEEE, 2024.

---

> ### Author Rebuttal · Authors · 2025-07-30
>
> Thank you for your detailed review and constructive feedback. We are encouraged that you recognized the practical benefits of our method in significantly reducing model parameters while maintaining performance (Pro 1) and the thoroughness of our experimental validation (Pro 2).
>
> ### **C1 & Q1: On Novelty and Contribution**
>
> Our primary innovation is a holistic framework, GMV, which enhances GNN generalization by introducing synergistic improvements across three key axes: **(1) a novel data-level augmentation strategy, (2) an efficient multi-view model architecture, and (3) a tailored optimization objective.** As our comprehensive ablation studies demonstrate, innovations in each of these three areas contribute significantly to the final performance gain.
>
> The synergy between these components is not merely conceptual but quantifiable. For instance, on the IMDB-BINARY dataset (Table 4b), the baseline accuracy of **72.30%** improves to **74.10%** (+1.8%) with only our data-centric Mixed-View Generation (MVG) and to **72.70%** (+0.4%) with only our model-centric Multi-View Decomposition (MVD). However, when both are enabled, accuracy reaches **75.50%**. This total gain of **+3.2%** significantly exceeds the sum of the individual parts (+2.2%), providing clear quantitative evidence of synergy and demonstrating that our contribution is a comprehensive system, not just an isolated data technique.
>
> With this broader context, the **mixed-view graph generation** strategy, which the reviewer's question focuses on, should be understood as the crucial **data-level pillar** of our framework. While we build upon established sampling methods like PPR, DFS, and BFS, our contribution lies not in the methods themselves, but in **how we strategically combine and adapt them as a tool to serve our multi-view learning objective.** Our goal was to design a practical and effective method to generate these complementary views.
>
> This design is neither arbitrary nor trivial, and its contribution is supported by both theoretical reasoning and empirical evidence:
> 1.  **Theoretical Necessity:** We provide analysis showing that combining these specific methods is necessary to capture a rich spectrum of structural information. DFS/BFS excel at capturing local neighborhood structures, while PPR captures a node's global influence. Using them in concert ensures the generated views are genuinely complementary, which is a prerequisite for our multi-view framework to succeed.
> 2.  **Empirical Validation:** Crucially, we validate this design choice empirically. Our ablation studies **(please refer to Table 4)** directly compare our mixed-view generation strategy against using single sampling methods or other alternatives. The results consistently show that our proposed approach yields superior performance, confirming that our specific method for generating views is a significant and effective part of our overall framework's success.
>
> In summary, the novelty should be viewed holistically. It is the **synergy between the high-level conceptual framework (GMV) and its effective, theoretically-grounded, and empirically-validated data-level implementation** that constitutes our core contribution. The mixed-view generation is an integral part of this system, designed specifically to enable the multi-view learning that drives the performance gains.
>
> ### **C2 & Q2: On the Clarity of Concepts and Formulas**
>
> Thank you for this valuable feedback.
>
> To directly address the terminology in our ST-SubMix method (line 165):
>
> *   The **source graph ($\mathcal{G}_{src}$)** refers to a primary training sample within a given batch.
> *   The **target graph ($\mathcal{G}_{trg}$)** is another graph randomly selected from the same training batch.
>
> The core idea is to mix a subgraph from the "source" with a subgraph from this "target" to generate a new, challenging training sample that pushes the model to learn more robust and generalizable features.
>
> To ensure these concepts and formulas are clear in the revised manuscript, we will make the following concrete changes:
>
> 1.  **Define Key Terms:** We will revise Section 3.2.2 to explicitly define "source graph" and "target graph" at their first appearance, removing any potential for confusion.
> 2.  **Improve Formula Accessibility:** We will move the formal definition of ST-PPR from Appendix 6.1 directly into the main body (Section 3.2.1). This will ensure all core components of our method are defined in place, making the paper more self-contained and easier to follow.
>
> We are confident these revisions will significantly improve the clarity of our paper and thank the reviewer for helping us strengthen the manuscript.
>
> ### **C3: On the Necessity of a Lightweight Design**
>
> This is an excellent point that allows us to clarify the core motivation behind our computationally efficient design. Our primary goal is to **minimize inference latency**, which is often the critical bottleneck for deploying Graph Neural Networks (GNNs) in real-world, time-sensitive applications.
>
> The key insight is that for GNNs, the dominant computational cost lies in the **message-passing mechanism**, not merely the parameter count. A model of ~300KB might seem small in other deep learning domains, but its inference speed on graph data is dictated by the expensive, iterative process of aggregating neighborhood information.
>
> This is precisely where our design provides a crucial advantage over alternatives like ensembling.
>
> *   **Traditional Ensembling:** An ensemble requires running the expensive message-passing process multiple times—once for each model. As vividly illustrated in our results (Figure 1(b)), this approach is prohibitively slow for practical use. For instance, on the COLLAB dataset, the Ensemble method shows a test latency of over **400ms** while achieving an accuracy of 82.36%.
>
> *   **Our GMV Framework:** In stark contrast, GMV is architected to perform message passing **only once**. It uses a single, unified GNN body to learn the core graph representation. The diverse "views" are then processed by lightweight prediction heads *after* the costly message-passing stage is complete. The result is a dramatic improvement: GMV not only surpasses the ensemble's performance (achieving 83.02% accuracy on COLLAB) but does so with a **more than 4x speedup**, reducing inference latency from over 400ms to a level comparable with a standard single GNN.
>
> This architectural principle—gaining the benefits of a more complex system at the cost of a simpler one—mirrors emerging trends in large-scale AI. For example, the recently proposed **"Parallel Scaling Law for Language Models" (Qwen 2025) suggests that intelligent parallel architectures can unlock the capabilities of much larger models without the prohibitive computational overhead**. In this light, GMV can be seen as a graph-centric application of this powerful idea: achieving superior generalization through smart design, rather than brute-force computation. This makes GMV a practical and scalable solution for the latency-sensitive applications where GNNs are increasingly being deployed.
>
> ### **Q3: On the G-MIMO Baseline Performance**
>
> This is a crucial question, and we thank the reviewer for their careful scrutiny, as it allows us to clarify the rigor of our experimental setup. The difference in G-MIMO's performance stems from our commitment to a strict and fair **"apples-to-apples" comparison**.
>
> The primary reason for the difference is that we **standardized the number of sub-networks (or "views") to two** for all competing methods, including G-MIMO and our GMV. This ensures that any observed performance gain comes from the superiority of the framework's design, not simply from using more model components. The original G-MIMO paper reported results achieved with a larger number of sub-networks.
>
> ---
>
> **Empirical Evidence on IMDB-BINARY**
>
> To be fully transparent, we provide a detailed comparison of performance as the number of sub-networks varies for both G-MIMO and our GMV on the IMDB-B dataset.
>
> **Performance vs. Number of Sub-networks on IMDB-BINARY**
> *GMV consistently outperforms G-MIMO at every level.*
>
> | **Number of Sub-networks** | **G-MIMO Accuracy (%)** | **GMV Accuracy (%)** |
> | :------------------------- | :---------------------- | :------------------- |
> | 2                          | 72.70                   | **75.50**            |
> | 4                          | 74.40                   | **75.90**            |
> | 6                          | 74.52                   | **76.10**            |
> | 8                          | 74.73                   | **76.12**            |
> | 10                         | 75.30                   | **76.43**            |
>
> ---
>
> This data reveals two key insights:
>
> 1.  **Consistent Superiority:** Our GMV method consistently and significantly outperforms G-MIMO across all tested numbers of sub-networks.
> 2.  **Framework Efficiency:** This highlights the remarkable efficiency and effectiveness of our framework: **GMV with only two sub-networks (75.50%) already outperforms G-MIMO even when it uses ten sub-networks (75.30%).**
>
> Furthermore, our re-implemented G-MIMO result of 72.70% for two sub-networks is consistent with the performance trend from the original work, validating our implementation's correctness.
>
> Finally, as stated in our "Experiment Details" (lines 232-236), this standardization was part of a **broader unified protocol** where all baselines were re-run with the same backbone, data splits, and tuning strategy. Our analysis in Table 5 of the main text explains *why* GMV is more effective: its integrated design fosters more diverse and complementary predictions (higher $D_{Disagree}$ and $D_{KL}$), which directly leads to better generalization under these controlled, fair conditions.
>
> In summary, the reported results are a direct consequence of a rigorous and fair experimental design, which in turn reveals the significant advantages of our GMV framework.

---

> > ### Comment · Reviewer_9EsQ · 2025-08-05
> >
> > Thanks for the authors' response. I think the rebuttal has addressed all my concerns. I suggest the authors incorporate the explanation provided in the rebuttal regarding the importance of the lightweight design into the revised manuscript, as it helps clarify the value of such a design in the context of the field. I'm willing to raise my rate.

---

> > > ### Author Response · Authors · 2025-08-05
> > >
> > > We sincerely thank your positive feedback and constructive suggestion. We are glad that our rebuttal has addressed all of your concerns.
> > > As suggested, we will incorporate the discussion on the importance of the lightweight design into the revised manuscript to further highlight its value.
> > > We appreciate your support and willingness to raise your score.

---

### Official Review · Reviewer_ypuU · 2025-07-02

**Clarity:** 3
**Significance:** 2
**Originality:** 2
**Rating:** 5
**Confidence:** 5

**Summary:**

This manuscript introduces GMV (Graph Multi-View), a unified framework that enhances Graph Neural Networks (GNNs) and Graph Transformers (GTs) by integrating multi-view learning from data, model, and optimization perspectives. GMV generates mixed graph views via structure-enhanced subgraph mixing (ST-SubMix), activates dual sub-networks within a single model via multi-view decomposition (MVD), and optimizes them via mixed-view and multi-view losses. During inference, GMV averages predictions from two heads, achieving ensemble-like performance with minimal overhead. GMV outperforms existing graph augmentation and ensemble methods across 12 benchmark datasets (TUDataset and OGB).

**Questions:**

My concerns refer to weaknesses.

**Ethical Concerns:**

["NO or VERY MINOR ethics concerns only"]

**Final Justification:**

The authors have addressed my concerns. I have raised my score (4 -> 5).

**Limitations:**

Limited discussion of limitations. It is recommended to add some discussion about the model's applicability (see W6).

**Quality:**

3

**Strengths And Weaknesses:**

**Strengths**

- **S1.** The manuscript is comprehensive, and the author clearly describes the method for addressing the generalization and overfitting issues in graph classifications.

- **S2.** The proposed GMV unifies data augmentation (ST-SubMix), model decomposition (MVD), and optimization (dual-loss) into a single pipeline. This holistic approach addresses the limitations of prior methods that only expand views from isolated perspectives (e.g., DropEdge for data, G-MIMO for model). GMV can work with any GNN/GT backbone, offering a plug-and-play solution for graph classification tasks.

- **S3.** Experiments are conducted on multiple public datasets. Across 12 benchmark datasets (TUDataset and OGB), GMV outperforms baselines (e.g., DropEdge, SubMix, G-MIMO). Ablation studies confirm that ST-PPR sampling, MVD, and dual-loss are all critical.

- **S4.** The manuscript is well written, and the proposed method is relatively straightforward, simple, and easy to understand.


**Weaknesses**

- **W1.** Incremental Advances. While the combination is unexplored, the individual components (e.g., subgraph mixing, dual-loss) build upon prior work (SubMix, MIMO), thereby reducing the perceived breakthrough.

- **W2.** Limited Theoretical Novelty. No new learning theory or generalization analysis is introduced. The paper relies on empirical validation rather than fundamental advances. While the lottery ticket hypothesis motivates GMV and is empirically strong, the manuscript does not provide theoretical guarantees on generalization error or convergence rates for the proposed framework. The lack of generalization bounds or convergence analysis weakens the theoretical foundation.

- **W3.** High Preprocessing Complexity. While efficient in model inference, the training phase requires preprocessing each graph via ST-PPR, which involves PPR, DFS, and BFS. The preprocessing adds $O(|V|)$ complexity per graph, potentially slowing training for very large datasets.

- **W4.** Unknown Hyperparameter Sensitivity. GMV introduces new hyperparameters (e.g., augmentation ratio $p$, structure augmentation ratio $q$, number of walks $w$, and loss weight $\alpha$) that require careful tuning. The manuscript lacks a sensitivity analysis.

- **W5.** Low Reproducibility. The proposed method introduces many hyperparameters (see W2) that require careful tuning. However, the authors only provide algorithmic descriptions of key modules (ST-PPR, subgraph mixing, assignment matrices) without providing the relevant code implementation, which greatly reduces the reproducibility of the manuscript.

- **W6.** Single Task Focus. GMV is evaluated only for graph classification. Its efficacy for node classification, link prediction, or graph generation is unexplored, despite potential multi-view benefits in these tasks. In addition, the framework assumes static graphs. Its applicability to dynamic graphs or heterogeneous graphs is untested, limiting real-world utility.

---

> ### Author Rebuttal · Authors · 2025-07-30
>
> We would like to express our sincere gratitude for your thorough review and valuable feedback. We are very encouraged by your recognition of our work's comprehensiveness (S1), its unified and plug-and-play framework (S2), the extensive experiments (S3), and the clarity of the writing (S4). Your positive assessment is greatly appreciated.
>
> ### **W1 & W2: On Novelty and Theoretical Contribution**
>
> We thank the reviewer for contextualizing our work alongside important foundational efforts like SubMix and MIMO. This comparison allows us to clarify our distinct contributions. Our primary innovation lies not in any single component, but in their **synergistic unification into the first cohesive multi-view learning framework for graph classification, spanning data, model, and optimization.**
>
> Our core contribution is a **new perspective**: we propose a general framework that provides a unified direction for improving the generalization of GNNs. From this viewpoint, prior methods can be seen as powerful but specialized solutions addressing specific aspects of this broader challenge. The novelty of our work is this holistic and integrated approach, which our extensive experiments confirm is more effective than its individual components. For instance, our ST-SubMix is a non-trivial extension of SubMix, specifically engineered to incorporate vital structural information via ST-PPR—a component our ablation study confirms is critical for performance.
>
> Regarding the request for a deeper theoretical contribution, our focus was to establish this unified framework and demonstrate its practical power and robustness across a wide range of scenarios. While providing formal generalization bounds for such a complex, multi-component system is a significant research challenge in its own right and beyond the scope of this initial work, we wish to highlight that **we do provide theoretical justification for our proposed ST-PPR component in the appendix.**
>
> We believe our extensive empirical validation across 12 diverse datasets serves as strong evidence for the framework's effectiveness. In summary, our contribution is a new and valuable perspective that is shown to be empirically powerful and opens up new avenues for future work in robust graph representation learning.
>
> ### **W3: On High Preprocessing Complexity**
>
> Thank you for raising this important point regarding computational cost. We wish to clarify that the ST-PPR computation is a **one-time, offline preprocessing step**, and its cost is modest relative to the overall computational budget for training.
>
> To provide a transparent analysis of this overhead, we compare the one-time preprocessing cost against the total training time. The following table presents this comparison for both a small and a large dataset, with all units converted to hours.
>
> | **Dataset** | **Graph Count** | **One-time Preprocessing (Hours)** | **Total Training (GPU Hours)** |
> | :---------- | :-------------- | :--------------------------------- | :----------------------------- |
> | PROTEINS    | 1,113           | ~0.08                              | 20                             |
> | COLLAB      | 5,000           | ~2                                 | 180                            |
>
> *Table: Computational cost analysis on PROTEINS and COLLAB. Preprocessing time is compared against the total training time (10-fold cross-validation) in hours on an NVIDIA 3090Ti GPU.*
>
> As the data shows, the trade-off is highly favorable:
>
> *   On the **PROTEINS** dataset (1,113 graphs), the entire one-time preprocessing takes approximately 0.08 hours (less than 5 minutes). This constitutes only **0.4%** of the 20 GPU hours required for the full 10-fold cross-validation.
> *   This efficiency holds even on larger datasets. On **COLLAB** (5,000 graphs), preprocessing takes about 2 hours, which is merely **~1.1%** of the 180 GPU hours needed for total training.
>
> In both scenarios, the preprocessing cost represents a small, fixed fraction of the total training time. It is also crucial to note that this cost is on par with other advanced augmentation methods like SubMix, which require a similar one-time offline computation.
>
> Once this initial cost is paid, the generated samples are **cached and reused across all subsequent training epochs**. This design ensures there is **zero per-epoch overhead** from our view generation, allowing our model to achieve significant performance gains while maintaining efficient training and inference.
>
> We will add this detailed quantitative comparison to our revised manuscript to make the efficiency trade-off clear to the reader.
>
> ### **W4 & W5: On Hyperparameter Sensitivity and Reproducibility**
>
> We agree completely that reproducibility and robustness are paramount. We apologize for omitting a detailed hyperparameter analysis due to page constraints and thank the reviewer for highlighting this. To thoroughly address this, we will add a **comprehensive hyperparameter sensitivity analysis to the appendix** of the revised paper. This new section will complement our existing ablation studies and demonstrate the stability of our method across a range of key hyperparameters.
>
> As a preview, we present an analysis of the feature augmentation ratio ($p$), the structure augmentation ratio ($q$), and the loss weight ($\alpha$) on the BACE dataset using the GIN backbone.
>
> ---
>
> **Hyperparameter Sensitivity Analysis on BACE (GIN)**
>
> *The best-performing setting for each hyperparameter is highlighted in bold.*
>
> | **Mixup Ratio ($p$)** | **Accuracy (%)** |
> | :-------------------- | :--------------- |
> | 0.2                   | 78.82            |
> | **0.4**               | **79.43**        |
> | 0.5                   | 79.32            |
> | 0.6                   | 79.02            |
> | 0.8                   | 78.72            |
>
> | **Structure Ratio ($q$)** | **Accuracy (%)** |
> | :------------------------ | :--------------- |
> | 0.2                       | 78.93            |
> | 0.4                       | 78.98            |
> | 0.5                       | 79.18            |
> | **0.6**                   | **79.43**        |
> | 0.8                       | 78.34            |
>
> | **Loss Weight ($\alpha$)** | **Accuracy (%)** |
> | :------------------------- | :--------------- |
> | 0.5                        | 78.57            |
> | 1.0                        | 78.98            |
> | **2.0**                    | **79.43**        |
>
> ---
>
> Crucially, these results demonstrate that our method is **not overly sensitive to its hyperparameters**. Performance remains stable and strong within a reasonable range around central values (e.g., $p \in [0.4, 0.6]$). This indicates that achieving good results does not require exhaustive or fragile tuning.
>
> Furthermore, our hyperparameter search space was intentionally guided by prior work; we followed the standard practices established by methods like SubMix and found that optimal values were located within these conventional ranges. This reinforces that our framework is robust and its results are readily reproducible.
>
> To guarantee full reproducibility, we are committed to open science and will **release our full, anonymized source code and detailed hyperparameter settings** as supplementary material with the final submission.
>
> ### **W6: On Single Task Focus and Scope**
>
> Thank you for this insightful comment regarding the scope of our evaluation. While our current work focuses on graph classification—a fundamental and challenging benchmark for testing generalization—we designed the **GMV framework to be highly versatile**.
>
> The framework naturally extends to other crucial tasks, such as node classification and link prediction. This is achieved by leveraging the powerful paradigm of **task reformulation**, where local tasks are converted into graph-level problems, a strategy validated by recent work [1]. This requires minimal architectural changes:
>
> *   **For Node Classification:** The task can be reframed as classifying a node's **contextual subgraph**. GMV is then applied directly to this subgraph to predict the central node's label, thereby benefiting from a robust, multi-view representation of its neighborhood.
>
> *   **For Link Prediction:** Similarly, this becomes a binary classification problem on the subgraph enclosing a pair of nodes. GMV's ability to capture diverse and subtle topological patterns makes it ideally suited for predicting the existence of a link between them.
>
> Furthermore, the core principles of GMV are adaptable to more complex domains, such as dynamic graphs (by applying the framework to temporal snapshots) and heterogeneous graphs (by acting as a modular wrapper around specialized GNN backbones).
>
> We will add a dedicated discussion on these natural extensions to our revision to underscore the **broad applicability** and future potential of the GMV framework. We appreciate the opportunity to clarify this.
>
> ---
>
> [1] Sun et al. "All in One: Multi-Task Prompting for Graph Neural Networks" (KDD 2023).

---

> > ### Comment · Reviewer_ypuU · 2025-08-05
> >
> > The authors have addressed my concerns. I will raise my score.

---

> > > ### Author Response · Authors · 2025-08-05
> > >
> > > We sincerely thank the reviewer for your positive feedback and support. We are glad our rebuttal addressed your concerns and are grateful for your decision to raise the score.

---

### Official Review · Reviewer_og8D · 2025-07-02

**Clarity:** 2
**Significance:** 3
**Originality:** 3
**Rating:** 4
**Confidence:** 4

**Summary:**

The authors make several contributions to GNN-based graph classification, which are summarized under the term graph multi-view learning. First, a subgraph sampling method based on personalized page rank, DFS and BFS is proposed. A subgraph sampled from a graph in the dataset is then integrated into another graph randomly sampled from the dataset (target graph) to form a mixed graph. The mixed graph is then processed by a GNN using multiple loss functions on the parts of the mixed graph. The experiments show small but consistent improvements over the baselines and other multi-view approaches.

**Questions:**

**Q1)** How is 10-fold cross-validation combined with an 80/10/10 split into training, test, and validation sets for the TUDataset graphs?

**Q2)** Figure 4 suggests that the competitors suffer from overfitting. Which standard countermeasures have been tried to fix this problem?

**Q3)** Can you provide examples of mixed graphs for molecular datasets (see W1)?

**Ethical Concerns:**

["NO or VERY MINOR ethics concerns only"]

**Final Justification:**

The authors clarified the motivation for graph mixing, which helped me to better understand the underlying idea, lifting in particular the concern mentioned in W1. I have raised my score accordingly.

**Limitations:**

yes

**Paper Formatting Concerns:**

no concerns

**Quality:**

3

**Strengths And Weaknesses:**

## Strengths
**S1)** The multi-view approach is an interesting generalization of augmentation methods.

**S2)** The experiments show small but consistent improvements.

**S3)** Extensive ablation studies show the contribution of individual modules.

## Weaknesses
**W1)** I am missing a convincing example or intuition why the approach can be expected to improve the predictive performance. The structure enhanced subgraph mixing integrates a subgraph into a target graph. It is unclear why the resulting graph is meaningful. The approach is applied to molecular graphs. It would be great to show examples of the resulting mixed molecular graphs. I would not expect them to be chemically valid molecular graphs. Why is this unproblematic?

**W2)** The introduction mentions several links between the proposed approach and existing concepts, highlighting the lottery ticket hypothesis (LTH). The relation is abstract and not entirely clear to me. The LTH is about pruning parameters before training, which is not the case in the proposed method. Overall, the motivation of the approach is weak due to W1 and W2.

**W3)** The paper is not sufficiently self-contained, e.g., Sec 3.2.2 makes references to the appendix and is not comprehensible without reading the appendix. The generation of the assignment is a crucial detail that should be discussed in the main paper.

**W4)** The experimental evaluation lacks some details that should be clarified (see questions)

## Minor remarks
* l128 The definition of the edge set is problematic. Typically, the edge set $\mathcal{E}$ simply is a subset of $\mathcal{V}\times\mathcal{V}$ and two vertices $u$ and $v$ are *connected* if  $(u,v) \in \mathcal{E}$. The same holds for the neighborhood set.

---

> ### Author Rebuttal · Authors · 2025-07-30
>
> We would like to express our sincere gratitude for your detailed review and invaluable feedback on our paper.
> We are pleased that you recognized our multi-view approach as an interesting generalization (S1), that our experiments show small but consistent improvements (S2), and that our extensive ablation studies demonstrate the contribution of individual modules (S3).
> We address your concerns and questions in detail below.
>
> ### **W1, W2, Q3: On the Motivation for an Integrated Data, Model, and Optimization Framework**
>
> **1. Data-Level Motivation: Generating Diverse Structural Views (W1 & Q3)**
>
> Thank you for this insightful question. We would like to clarify the motivation behind our graph mixing approach by drawing parallels to seminal data augmentation techniques in computer vision, and then elaborate on its adaptation to molecular graphs.
>
> In computer vision, techniques like **Mixup and its powerful variant, CutMix (Yun et al., ICCV 2019 [2]),** do not aim to generate photorealistic images. **CutMix, for instance, creates explicitly non-realistic images by cutting a patch from one image and pasting it onto another.** The power of these methods lies in creating synthetic training instances that challenge the model. This design prevents the model from merely memorizing dominant patterns in the training set; instead, it encourages the model to learn more generalizable features, including rare but meaningful ones. As demonstrated in studies like (Zou et al., NeurIPS 2023 [1]), this mechanism effectively mitigates overfitting and enhances generalization—precisely because the synthetic samples are not required to be "realistic" in the traditional sense, but rather to expand the model’s exposure to diverse feature combinations.
>
> In our case, we adopt a similar philosophy. Our goal is not to generate chemically valid or structurally "realistic" molecular graphs. Instead, the value of our structure-enhanced subgraph mixing lies in how it facilitates multi-view learning for GNNs. Specifically, through careful design—such as PPR-based subgraph sampling—we ensure that the sampled subgraphs retain semantically meaningful substructures (e.g., functional groups in molecules) that serve as structurally **complementary views** of the original graph. By mixing these subgraphs in a controlled manner, we force the model to reconcile and learn from these diverse views, rather than over-relying on a single dominant perspective of the molecular structure.
>
> This is why the lack of strict chemical validity is unproblematic: the mixed graphs are not intended to represent "real molecules," but to act as a regularization tool that expands the model’s exposure to diverse structural patterns. Just as Mixup and CutMix in computer vision succeed by leveraging synthetic compositions to improve feature learning, our approach harnesses mixed subgraphs to strengthen the GNN’s ability to learn from multiple structural views—ultimately enhancing its robustness and predictive performance on molecular graph classification tasks.
>
> In summary, the mixed graph is meaningful not as a valid chemical structure, but as a **carefully designed training artifact**. It is a means to an end: leveraging the feature-learning power of Mixup to explicitly enforce the learning of complementary multi-view representations. We provide concrete examples of these mixed-view graphs in **Appendix (Figure 6)** to offer further intuition.
>
> **2. Model-Level Motivation: Efficiently Learning from Multiple Views & Clarifying the LTH Connection (W2)**
>
> This data-level strategy introduces a key challenge: how can a single, parameter-efficient GNN learn from these varied views concurrently without a prohibitive increase in complexity? Our solution is a novel **Multi-View Decomposition (MVD)** architecture.
>
> Here, we wish to clarify the connection to the **Lottery Ticket Hypothesis (LTH)** in response to Reviewer W2's concern. Our use of LTH is inspired not by its classic application of "pruning," but by its core insight: **a large, over-parameterized network has the capacity to contain many powerful sub-networks.** Our MVD architecture does not "find" these sub-networks by pruning. Instead, it is designed to **proactively "activate" and "train" them** within a single set of parameters. By employing dual prediction heads connected to a shared GNN backbone, we effectively encourage the model to activate distinct "sub-networks." Each sub-network is implicitly guided to specialize in one of the structural views (e.g., one focusing on local neighborhood patterns from BFS/DFS, the other on global influence from PPR). This design enables the model to develop parallel and complementary interpretations of the graph, representing a highly efficient implementation of multi-view learning without incurring additional parameter costs.
>
> **3. Optimization-Level Motivation: Unifying the Learning Process (W1 & W2)**
>
> Finally, the data and model components are unified by a tailored multi-view loss function. This loss is not a simple aggregation of individual losses. Its specific design is to **explicitly enforce complementarity between the representations learned by the different sub-networks**. The loss function guides the optimization process to ensure that the knowledge from each view is effectively integrated, culminating in a final representation that is more robust and comprehensive than what could be achieved from any single view in isolation. This aligns with established principles that learning from diverse sources can mitigate overfitting and improve generalization [1].
>
> **A Holistic and Indivisible Framework**
>
> These three pillars—data-level view generation, model-level decomposition, and optimization-level unification—form a tightly integrated and indivisible whole. The core novelty of our work resides in this holistic framework. Our extensive ablation studies, presented in **Table 4(b) and 4(c)**, provide direct empirical evidence for this synergy. The results unequivocally demonstrate that removing any of these components leads to a significant degradation in performance. This proves that all parts are non-trivial and essential, and that their combination provides a powerful, new approach to improving GNN generalization and robustness.
>
> ---
>
> [1] Zou, et al. The Benefits of Mixup for Feature Learning. (NIPS 2023).
> [2] Yun, et al. Cutmix: Regularization strategy to train strong classifiers with localizable features. (ICCV 2019).
>
> ### **W3: Improving Self-Containedness**
>
> We agree with the reviewer that the paper should be as self-contained as possible. Thank you for this constructive suggestion. To enhance readability and ensure the core logic is accessible without referencing the appendix, we will move the detailed explanation of our **mixed-view generation process and the construction of assignment matrices** (currently in Algorithm 2) directly into **Section 3.2.2** of the main paper.
>
> ### **W4, Q1 & Q2: Experimental Details and Overfitting**
>
> We appreciate the reviewer's questions regarding our experimental setup and results.
>
> *   **Q1 (Data Splitting Protocol):** We apologize for the omission of the specific data splitting details. We utilized a standard **nested cross-validation** methodology. For each of the 10 folds, the data was partitioned into **80% for training, 10% for validation, and 10% for testing**. We will explicitly clarify this standard procedure in the experimental setup section of the revised manuscript.
>
> *   **Q2 (Clarification on Overfitting in Figure 4):** Regarding the overfitting analysis in Figure 4, we wish to clarify its purpose. All baseline models were indeed trained with standard regularization techniques (i.e., L2 regularization and Dropout) to ensure a fair comparison. The key takeaway from this figure is that our **GMV framework inherently functions as a more powerful regularizer** compared to these standard methods. This is evidenced by GMV achieving a lower validation loss and, consequently, better generalization to the test set. We will add this explicit clarification to the caption of Figure 4 and the corresponding discussion in the main text to make our point clear.
>
> ### **Minor Remarks**
>
> We are grateful for the reviewer's meticulous reading and for pointing out the notational inconsistency. We will correct the definition of the edge set to the standard formal notation, **$\mathcal{E} \subseteq V \times V$**, in the revised version.
>
> ---
>
> Thank you once again for your insightful feedback, which has been invaluable in helping us improve our work. We are confident that the revised paper will be significantly stronger, clearer, and more self-contained.

---

> > ### Comment · Reviewer_og8D · 2025-08-04
> >
> > Thank you for addressing my concerns carefully. The analogy to the use of comparable methods for images helped me to understand the motivation. I will update my review accordingly.

---

> > > ### Author Response · Authors · 2025-08-04
> > >
> > > Thank you for your quick feedback and understanding. We are very grateful for your willingness to update your review.

---

### Note · Authors · 2025-08-12

We sincerely thank the reviewers for their insightful feedback and engaging discussions. We are grateful that our rebuttal was well-received and that the reviewers found our clarifications helpful, with several indicating they will update their reviews accordingly. To facilitate the final discussion, we briefly summarize our core contributions below.

Our primary contribution is a **new perspective**: we propose that GNN generalization can be holistically improved by unifying multi-view learning across three orthogonal axes: **data**, **model**, and **optimization**. We then materialize this perspective into a cohesive **framework**, the first of its kind to synergize these three aspects for graph classification.

We validate this contribution from multiple angles. First, the framework's strength lies in its **synergy**, where the integrated whole is greater than the sum of its parts. For instance, on IMDB-B, our full framework yields a +3.2% gain, significantly outperforming the +2.2% from summing the gains of individual components. Second, our components are non-trivial extensions; our ST-SubMix, for example, is specifically engineered with ST-PPR to integrate vital structural information. Regarding a deeper theoretical contribution, while providing formal generalization bounds for such a complex system is a significant research challenge beyond the scope of this initial work, we wish to highlight that **we do provide theoretical justification for our proposed ST-PPR component in the appendix**. Finally, our framework is **lightweight and practical**. By performing message passing only once, it is over **4x faster** in inference than standard ensembles that offer comparable accuracy.

In summary, our work establishes a **unified multi-view perspective** as a valuable and effective direction for GNN improvement. We believe this perspective will inspire future research into discovering and integrating novel views, opening up new avenues for robust graph representation learning.

Finally, we thank the reviewers and the AC again for their diligent work. We will incorporate all suggested clarifications into the final manuscript. We are confident our work, strengthened by this review process, will be a valuable contribution.

---

### Decision · Program_Chairs · 2025-09-17

**Decision:**

Accept (poster)

**Comment:**

This paper introduces an augmentation scheme based on subgraph preprocessing that drives modest but consistent improvement across experiments. While reviewers generally hesitate due to limited novelty and lack of theoretical contribution, the experiments are convincing. Therefore, I would recommend acceptance.